# Spectral Efficiency Maximization for Mixed-Structure Cognitive Radio Hybrid Wideband Millimeter-Wave Transceivers in Relay-Assisted Multi-User Multiple-Input Multiple-Output Systems

**DOI:** 10.3390/s24123713

**Published:** 2024-06-07

**Authors:** Hafiz Muhammad Tahir Mustafa, Jung-In Baik, Hyoung-Kyu Song, Muhammad Adnan, Waqar Majeed Awan

**Affiliations:** 1Department of Information and Communication Engineering, Sejong University, Seoul 05006, Republic of Korea; mustafasurrey@gmail.com (H.M.T.M.); bji0309@gmail.com (J.-I.B.); 2Department of Convergence Engineering for Intelligent Drone, Sejong University, Seoul 05006, Republic of Korea; 3Department of Computer Science (SST), University of Management and Technology, Lahore 54770, Pakistan; muhammad.adnan@umt.edu.pk; 4Department of Electrical Engineering, University of Management and Technology, Lahore 54770, Pakistan; f2017179001@umt.edu.pk

**Keywords:** cognitive radio, decode-and-forward relay, hybrid wideband transceiver, millimeter-wave technology, MU-MIMO system, spectral efficiency maximization

## Abstract

This paper proposes a cognitive radio network (CRN)-based hybrid wideband precoding for maximizing spectral efficiency in millimeter-wave relay-assisted multi-user (MU) multiple-input multiple-output (MIMO) systems. The underlying problem is NP-hard and non-convex due to the joint optimization of hybrid processing components and the constant amplitude constraint imposed by the analog beamformer in the radio frequency (RF) domain. Furthermore, the analog beamforming solution common to all sub-carriers adds another layer of design complexity. Two hybrid beamforming architectures, i.e., mixed and fully connected ones, are taken into account to tackle this problem, considering the decode-and-forward (DF) relay node. To reduce the complexity of the original optimization problem, an attempt is made to decompose it into sub-problems. Leveraging this, each sub-problem is addressed by following a decoupled design methodology. The phase-only beamforming solution is derived to maximize the sum of spectral efficiency, while digital baseband processing components are designed to keep interference within a predefined limit. Computer simulations are conducted by changing system parameters under different accuracy levels of channel-state information (CSI), and the obtained results demonstrate the effectiveness of the proposed technique. Additionally, the mixed structure shows better energy efficiency performance compared to its counterparts and outperforms benchmarks.

## 1. Introduction

Future wideband mobile wireless networks need to accommodate the explosively growing demand for ultra-high-speed data transmission. This requirement leads to the exploration of the millimeter-wave (mm-wave) frequency band (30–300 GHz), owing to its advantage of containing huge spectral resources [1]. Furthermore, mm-wave technology with large-scale multi-input multi-output (MIMO) (also referred to as massive MIMO) systems are two key findings to address the spectrum crunch and enhance spectral and energy efficiencies [2]. Therefore, it is possible to realize several newly emerging applications, such as smart environments, autonomous vehicles, industrial and agriculture automation, remote healthcare, high-quality three-dimensional (3D) video, holographic imaging, virtual reality, and augmented reality [3,4]. These applications essentially demand low latency, higher data rates, and enhanced connectivity. Additionally, innovative signal processing techniques are required to make mm-wave communication feasible on commercial grounds [5]. There are several reasons behind that, such as new hardware constraints, operations at higher frequencies and large bandwidths, mm-wave channel impairments, advanced circuit implementation, and power consumption challenges to deploy large-scale antenna arrays to attain high beamforming gain [6].

In the forthcoming wireless systems (e.g., beyond the 5th generation (B5G) networks), the inevitable proliferation of connected devices may lead to spectral congestion. Therefore, efficient utilization of available frequency resources is required to avoid this situation [7,8]. It was observed in the measurement campaign that conventional fixed license-based static spectrum allocation policies decrease spectrum utilization efficiency to a great extent [9,10,11]. To resolve this issue, a spectrum-sharing technique is suggested to enhance the efficiency of spectrum utilization, and this sharing mechanism is based on dynamic spectrum allocation policies [12,13]. Cognitive radio (CR) is a feasible solution that makes it possible to support dynamic spectrum access, allowing secondary users (SUs) to utilize licensed spectrum while ensuring the quality of service (QoS) of primary users (PUs) [14]. Just as minimizing the impact of interference on PUs caused by the SUs is important, it is also desirable to minimize the effect of interference on the SU due to the PU in CR communication. This task can be achieved by regulating the power transmission at the respective unit [15,16,17]. To maximize spectrum utilization in B5G wireless networks, it seems crucial to combine the dynamic spectrum allocation strategy with mm-wave transmission. Specifically, novel applications from diverse fields, such as wireless backhaul, the Internet of Things (IoT), and intelligent transportation systems, can be realized by exploiting the adaptability of CR technology combined with the advantages of mm-wave frequencies [18].

It is worth noting that the characteristics of mm-wave channels are different compared to the lower part of the frequency spectrum due to the smaller wavelength of transmitted signals. For instance, huge path loss, limited propagation paths, lower diffraction, and highly directional wireless channels are associated with mm-wave frequencies [19]. To overcome the poor characteristics of mm-wave channels, the combination of large-scale MIMO and hybrid beamforming is considered the most attractive solution [20]. In particular, massive MIMO can achieve significant beamforming gain, while hybrid processing reduces hardware complexity, cost, and power consumption by significantly reducing the number of RF chains. However, hybrid precoding design in practical wideband systems is a challenging task, as it requires the analog RF precoder and combiner common to all sub-carriers [21]. In mm-wave transmission, there is a definite need to address the blockage sensitivity of signals. This impairment makes it quite difficult to establish a reliable non-line-of-sight (NLOS) communication link due to excessive path loss [22]. Fortunately, relay-assisted transmission can facilitate decreasing transmission power and enhance network coverage, link quality, and communication range [23,24]. In conclusion, relay-assisted MIMO communication networks improve the overall performance, especially at the edge of the cell.

It is worth highlighting that conventional fully digital beamforming results in prohibitive hardware complexity and cost for large-scale MIMO systems, as this precoding demands the same number of RF chains as the number of antennas in an array [25,26]. Among other alternatives, hybrid beamforming has been widely investigated to provide a good trade-off between complexity and performance. In a hybrid architecture, the beamforming process is divided into the digital baseband and the analog RF domains. This precoding strategy significantly decreases the number of RF chains due to the low-dimensional digital baseband processing components [27]. Moreover, multiplexing gain is achieved through the baseband processing part, while beamforming gain is obtained via the RF processing unit. Note that a network of phase shifters can be employed for the practical implementation of the analog RF beamformer [28]. In general, hybrid beamforming architectures can be categorized as (1) a fully connected structure where all the RF chains are connected to each antenna in an array—this structure enables each RF chain to achieve its full beamforming gain—and (2) a partially connected structure where each RF chain is connected to a non-overlapping subset of the antennas [29]. A sub-connected architecture uses less phase shifters for a given number of antennas and RF chains in comparison to a fully connected one. Therefore, the partially connected architecture reduces hardware implementation complexity and provides a cost-effective and energy-efficient hybrid precoding solution, but at the cost of reducing some beamforming gain [30]. Contrary to the fully connected structure, it is challenging to sustain fine control over the beam with the sub-connected structure, which can lead to less accurate beamforming. Furthermore, the partially connected architecture limits the flexibility of large antenna arrays, which may cause a significant reduction in performance [31,32]. It is worth noting that orthogonal frequency division multiplexing (OFDM) can support substantially higher data rates over a wide frequency range, making it a suitable choice for CRN. Hence, it is required to design a hybrid broadband mm-wave transceiver for the CRN-based MU-MIMO OFDM systems. Also, an energy-efficient solution can be obtained using a mixed hybrid structure rather than a fully connected one. To derive efficient hybrid processing components at different communicating nodes by taking interference at the PU into consideration is quite a challenging aspect of this problem.

### 1.1. Related Work

The works in [33,34,35,36,37,38,39] illustrate that the highly directional beamforming capability of mm-wave technology can improve user data rates and effectively mitigate interference simultaneously. From the perspective of spectrum sharing, these features of mm-wave transmission are quite useful. The authors of [33] analytically prove that license sharing among operators can enhance the performance of mm-wave systems. In comparison to the exclusive license model, the authors in [34] demonstrate that performance gains might be increased by up to 130% using spectrum-sharing techniques. An optimization framework is proposed in [36] for mm-wave CRN by incorporating joint beamforming to maximize the capacity and fairness of users, base-station (BS) association, and coordination. In [39], the authors examine the performance of spectrum-sharing systems in the 26 GHz and 70 GHz frequency bands, considering both ideal and non-ideal beamforming approaches in a real propagation environment. Based on their research, a spectrum-sharing system can achieve a capacity that is two to three times higher than an exclusive licensing network. Nevertheless, these advantages are unlikely to be achieved due to beamforming errors and poor interference mitigation techniques, particularly in situations with a low signal-to-interference plus noise ratio (SINR). The authors in [40] suggest a technique for interference management in a spectrum-sharing network, and its viability is also proved through computer simulations.

Hybrid precoding techniques have been extensively examined in both single-user (SU) and MU-MIMO communication networks. In [41,42,43,44,45,46,47,48,49,50,51], the authors propose hybrid beamforming algorithms for wideband mm-wave MIMO systems. In [41], the authors develop a principal component analysis (PCA)-based hybrid broadband beamforming algorithm, where a frequency-selective fully digital precoder is employed to design the frequency-independent analog RF beamformer. Additionally, this work is restricted to SU-MIMO-OFDM networks. The authors in [42] develop a hybrid precoding method for mm-wave SU-MIMO and MU multi-input single-output (MU-MISO) systems. This technique can achieve performance equal to the corresponding fully digital beamforming, provided the number of RF chains doubles the number of transmitted data streams. In [43], the authors propose the OFDM-based hybrid transceiver for massive MIMO systems, and this work is an extension of the previous algorithm in [42] to make it compatible with practical broadband systems. Based on statistical channel information, a practical, low-complexity hybrid precoding technique is suggested in [47], which employs an efficient searching algorithm to select the best possible beamforming vectors from codebooks. The constrained Tucker2 tensor decomposition technique is used in [48] to design the hybrid broadband transceiver for mm-wave MU-MIMO communication networks. Furthermore, the focus of this scheme is to enhance the capacity of the equivalent baseband channels by minimizing the impact of inter-user interference (IUI). In [50], the authors exploit an alternating minimization algorithm for phase extraction to approximate the full-complexity matrix from the corresponding analog processing component, considering mm-wave MU-MIMO communication networks. The authors of [51] propose alternating minimization (Alt-Min) algorithms to address hybrid beamforming for mm-wave MIMO systems using frequency-selective channels. The authors in [52] suggest the orthogonal matching pursuit (OMP) algorithm for hybrid precoding design by assuming frequency-flat channels. Moreover, this technique exploits the sparsity of mm-wave channels to derive the analog RF and digital baseband processing components.

In [53,54], the authors investigate relay-assisted hybrid precoding for maximizing spectral efficiency in single-user and MU mm-wave massive MIMO networks with a fully connected structure. Moreover, the amplify-and-forward (AF) relaying protocol is considered at the relay station in these designs. The works in [55,56] also focus on relay-based hybrid beamforming for maximizing energy efficiency in single-user mm-wave large-scale MIMO systems using partially connected structures at communicating nodes. Specifically, iterative successive approximation (ISA) and alternating direction method of multiplier (ADMM) algorithms are used to achieve the desired goals in [50,55], respectively. In [57], the authors examine the hybrid transceiver for multi-hop AF relay systems by taking channel errors into account. A mixed hybrid architecture, comprising both partially connected and fully connected structures, is proposed in [58] based on matrix factorization to achieve near-optimal performance. This hybrid precoding strategy shows lower hardware complexity and higher energy efficiency when compared with the corresponding fully connected structure. The authors of [59] present a codebook-based generalized sub-array connected (GSAC) structure for hybrid transceiver design to enhance energy efficiency. The authors in [60] design an analog beamformer that aims to minimize the channel estimation error by leveraging the channel’s long-term properties. Then, the baseband processing matrix is computed by employing simple digital signal processing such as maximum ratio combining/maximum ratio transmission (MRC/MRT). In [61], hybrid beamforming for mm-wave MIMO relay networks is suggested using imperfect CSI, where the AF protocol is adopted at relay nodes. Moreover, both algorithms [60,61] consider fully connected hybrid structures.

Previous works show that substantial research has been conducted on spectrum sharing in mm-wave MIMO systems and hybrid beamforming design in mm-wave MIMO OFDM networks. Nevertheless, hybrid transceivers that integrate the concept of CRN have not been studied extensively. The authors of [62] attempt to leverage spectrum sharing for designing hybrid transceivers by taking backhaul networks into consideration. The hybrid beamforming techniques in [63,64] primarily focus on maximizing the minimum secrecy rate of all SUs while accounting for practical limitations. In [65], the authors propose MU underlay cognitive hybrid transceiver designs while keeping interference to incumbent users within a predefined threshold. This technique is also applicable to both the downlink and uplink, provided that complete CSI is available.

The authors of [54,66,67,68,69,70] primarily focus their attention on designing mm-wave relay-based hybrid precoding for MU-MIMO systems. In [66], the authors investigate the hybrid beamforming technique for sum rate maximization with mixed and fully connected structures using the DF relaying protocol. This design does not support frequency-selective channels, as the RF and baseband processing matrices are derived under narrowband assumptions. The authors in [67] also develop the fully connected hybrid beamforming algorithm, where the coordinated beam alignment procedure is adopted for designing the RF processing matrices. Moreover, a digital baseband solution is obtained using a non-linear precoding method. In [68], the authors also suggest the hybrid precoding method for mm-wave massive MU-MIMO DF relay systems, taking mixed structure into account. This scheme is not suitable for wideband systems, as the analog and digital beamforming components are derived considering frequency-flat fading channels. Moreover, this algorithm shows relatively better performance when compared with the presented scheme in [66]. The authors in [69] propose the hybrid transceiver for mm-wave massive MU-MIMO relay-assisted networks with partially connected structures, considering the AF relaying principle. Just like [66,68], this technique suffers from the same limitations. In particular, the hybrid precoding techniques presented in [66,67,68,69] are not applicable when cognitive radio comes into play. The algorithm proposed in [70] is compatible with CR-based fully connected hybrid transceivers, but there is a definite need to reduce cost, energy, and hardware complexity without significant performance degradation. This work is an attempt to address this gap, considering the mixed hybrid structure to achieve the desired goal. The distinguishing feature of this work compared to similar existing techniques [54,66,67,68,69,70] is illustrated in Table 1.

### 1.2. Motivation and Contribution

Contrary to the hybrid transceiver designs reported in the literature for relay-assisted MU-MIMO systems [54,66,67,68,69,70], there does not exist any CRN-based hybrid wideband mm-wave beamforming scheme that attempts to maximize the sum spectral efficiency, exploiting a low-complexity and energy-efficient structure. This provides motivation for the proposed hybrid transceiver design. To achieve the desired goal, the presented technique considers a relay-assisted MU-MIMO with a mixed structure, which comprises a combination of partially connected and fully connected architectures. This structure leads to a more practical solution that enhances energy efficiency when compared with the corresponding fully connected architecture. Additionally, the suggested mixed structure achieves sum spectral efficiency close to fully connected hybrid precoding with relatively low hardware complexity. Finally, the underlying hybrid beamforming design also leads to the efficient utilization of spectral resources, as the concept of CRN is also included in the proposed algorithm. The main contribution of this paper is summarized as follows:We investigate the hybrid broadband transceiver for sum spectral efficiency maximization in mm-wave relay-assisted MU-MIMO systems by incorporating the idea of CR technology to avoid inefficient spectrum allocation. The proposed algorithm primarily considers the mixed hybrid structure at the relay node and the partially connected architecture at each SU. This hybrid beamforming problem does not exist in the present literature to the best of the authors’ knowledge. Furthermore, the scenario investigated in this paper is consistent with practical considerations, as the underlying network architecture attempts to utilize the spectral resources efficiently while maximizing the sum rate using a low-complexity and energy-efficient hybrid structure, taking the transmitted power and interference constraints into account.In contrast to the existing hybrid precoding techniques for relay-assisted MU-MIMO systems [54,66,67,68,69,70], the proposed design considers frequency-selective channels, the CR paradigm, and mixed structures simultaneously, which offers great potential for unexplored applications. It is worth mentioning that the formulated optimization problem is non-convex due to the element-wise constant amplitude constraints imposed by the analog processing components. Also, the joint optimization of several complex matrix variables makes the original optimization problem NP-hard. Furthermore, it is required to derive the RF beamforming solution common to all sub-carriers. Consequently, design complexity and the associated computational complexity become prohibitively high. Therefore, the solution to the problem at hand is more challenging.To reduce the complexity of the optimization problem and make it tractable, the proposed algorithm first attempts to decompose the challenging task into two single-hop sum rate maximization sub-problems by exploiting the notion of information theory and the structural characteristics of DF relays. One sub-problem aims to maximize the sum rate from the source to relay decoding, while the other focuses on maximizing the sum spectral efficiency from relay encoding to multiple SUs. Then, a decoupled design approach is followed to drive the analog RF and digital baseband processing components. In particular, the derivation of RF processing components in each sub-problem attempts to maximize the sum rate, while digital baseband processing matrices focus their attention on minimizing interference. Furthermore, the mixed hybrid architecture at the relay node and the partially connected structure at each SU lead to an energy-efficient and low-complexity solution.The spatial wideband effect on the performance of the proposed scheme is also described through computer simulations. Furthermore, the impact of a uniform circular array under high-frequency transmission is also demonstrated using simulation results. Additionally, the proposed scheme also shows less computational complexity when compared with several other existing techniques.Numerical results are generated by changing system parameters over a wide range for both mixed and fully connected structures. Also, the performance curves are obtained under imperfect channel state information (CSI), considering the mixed hybrid architecture. It is evident from the simulation results that the suggested fully connected structure achieves performance close to full-complexity digital precoding, while the mixed hybrid structure shows relatively lower performance compared to the fully connected one. Furthermore, minor degradation in performance occurs when channel estimation error increases gradually. The obtained results also show that the proposed mixed architecture achieves significantly higher energy efficiency when compared to its fully connected counterpart. Finally, the achieved spectral efficiency and energy efficiency demonstrate the effectiveness of the proposed approach.

The rest of the paper is structured as follows: Section 2 describes the system model, the frequency domain mm-wave channel model under consideration, and problem formulation. Section 3 delineates the proposed hybrid transceiver design. The complexity analysis of the suggested technique is briefly discussed in Section 4, where a comparison is also made with other hybrid precoding algorithms. Section 5 presents simulation results for the performance evaluation, and concluding remarks are given in Section 6.

Notation: Upper-case and lower-case boldface letters denote matrices and vectors, respectively. $A^{H}$, $A^{T}$, ∡A, ${\left\|A\right\|}_{F}$, |A(i,j)|, A(:,i), and A(:,1:j) represent the conjugate transpose, transpose, element-wise phase, Frobenius norm, element-wise modulus, i-th column, and first j columns of a matrix A, respectively. ℂ depicts the field of complex numbers, ℝ specifies the field of real numbers, $I_{m}$ shows the identity matrix of order m×m, and $CN\left(0,\;\sigma^{2}I_{n}\right)$ describes the complex Gaussian distribution with mean 0 and covariance matrix $\sigma^{2}I_{n}$. The determinant and trace of a matrix A are given as det(A) and Tr(A), respectively. E[.] shows the expectation operator, and $bd\left(A_{1},\ldots ,A_{K}\right)$ stands for a block diagonal matrix with sub-matrices $A_{1},\ldots ,A_{K}$. Table 2 shows the summary of the main symbols.

## 2. System Model and Problem Formulation

This section describes a system model of the underlying hybrid transceiver with two different structures: (1) a fully connected structure and (2) a mixed architecture. Furthermore, the frequency-domain mm-wave channel model and problem formulation are also given in this section.

### 2.1. System Model

Consider a downlink of the CRN-based relay-assisted MU-MIMO system with hybrid precoding as illustrated in Figure 1, where the source node, relay station, and the k-th SU are equipped with $N_{t}$, $N_{r}$, and $N_{d_{k}}$ antennas, respectively. The number of RF chains installed at the above-mentioned communicating nodes is represented as $N_{t}^{RF}$, $N_{r}^{RF}$ and $N_{d_{k}}^{RF}$, respectively. The source node transmits data streams to K SUs through the relay node, while a direct communication link is employed for the PU. It is assumed that the direct transmission link between the source and K SUs is not favorable owing to excessive path loss and deep fading. To transmit $N_{s}$ data streams from the cognitive radio base station (CRBS) to each SU, it is necessary for the source and relay to handle $KN_{s}$ data streams. This requirement leads to the essential conditions $KN_{s}\leq \min\left(N_{t}^{RF},\;N_{r}^{RF}\right)\ll \min\left(N_{t},\;N_{r}\right)$ and $N_{s}\leq \min\left(N_{d_{k}}^{RF}\right)\ll \min\left(N_{d_{k}}\right)$. These conditions enable efficient multi-stream transmission using a significantly small number of RF chains. To avoid relay-induced signal-space collisions, another condition $N_{t}\geq \;N_{r}\geq \sum_{k=1}^{K}N_{d_{k}}$ also needs to be satisfied.

In relay-based MIMO networks, two time slots are required to complete the transmission from the source to the destination. In the first time slot, the data streams of all intended users are transmitted from the source to the relay node. In the second time slot, the received signal at the relay station is processed and then forwarded to the end users. From this perspective, the communication link from the CRBS to the SUs can be decomposed into a single-user MIMO system (from the source to the relay station) and a MU-MIMO system (from the relay node to multiple users).

Let sk[n]∈ℂNs×1 be a vector of complex data streams intended to the k-th SU such that E[sk[n]skH[n]]=INs,∀n∈{1,…,Nsub}, and let s[n]∈ℂKNs×1 be a vector of complex information symbols to K SUs, which can be written as s[n]=[s1T[n],…,sKT[n]]T, where n denotes the sub-carrier index, and Nsub specifies the total number of sub-carriers. At the CRBS, the frequency-selective digital baseband beamformer VBB[n]=[VBB,1[n],…,VBB,K[n]]∈ℂNtRF×KNs is first employed to precode KNs data streams, specified by vector s[n]. Then, the frequency-flat analog RF beamformer VRF∈ℂNt×NtRF is deployed to enhance the beamforming gain. The transmitted signal x[n]∈ℂNt×1 from the source at the n-th sub-carrier is given as x[n]=VRFVBB[n]s[n]=VRF∑k=1KVBB,k[n]sk[n]. Considering the transmit power constraint at the source, we have E[x[n]xH[n]]=Tr{(VRFVBB [n])(VRFVBB [n])H}≤Ps, ∀n∈{1,…,Nsub}, where Ps indicates the maximum transmit power.

Note that VRF is common to all Nsub sub-carriers. Therefore, the frequency-independent analog processing component needs to be derived for hybrid wideband transceivers. The signal received yr[n]∈ℂNr×1 at the relay station can be modeled as
(1)yr[n]=H[n]VRF∑k=1KVBB,k[n]sk[n]+nr[n], 
where H[n]∈ℂNr×Nt represents the channel matrix from the source to relay station in frequency-domain, and nr[n]∈ℂNr×1 is the zero mean circularly symmetric complex Gaussian (ZMCSCG) noise with variance σr2, i.e., nr[n]~CN(0, σr2INr). Due to the source-transmitted signal x[n]∈ℂNt×1, the interference experienced by the PU can be characterized as
(2)d1[n]=‖HPU[n]VRF∑k=1KVBB,k[n]‖F2, 
where HPU[n] denotes the channel matrix from the source to the PU at the n-th sub-carrier. The equivalent baseband signal yr1[n]∈ℂKNs×1 received at the output of relay hybrid combiner F1[n]=FRF,1FBB,1[n]∈ℂNr×KNs can be expressed as
(3)yr1[n]=(FRF,1FBB,1[n])H(H[n]VRF∑k=1KVBB,k[n]sk[n]+nr[n]),
where FRF,1∈ℂNr×NrRF and FBB,1[n]∈ℂNrRF×KNs are the frequency-flat analog combiner and the frequency-dependent digital baseband combiner at the relay node, respectively. Applying hybrid beamforming, the transmitted signal yr2[n]∈ℂNr×1 from the relay node is given as
(4)yr2[n]=FRF,2FBB,2[n]yr1[n]=(FRF,2FBB,2[n])(FRF,1FBB,1[n])H︸relay hybrid filter (H[n]VRF∑k=1KVBB,k[n]sk[n]+nr[n]),
where the relay node is supposed to apply frequency-dependent digital baseband precoder FBB,2[n]∈ℂNrRF×KNs followed by the frequency-independent analog RF beamformer FRF,2∈ℂNr×NrRF. It is worth highlighting that the partially connected RF processing component in hybrid precoding and combining can further reduce the cost, energy, and hardware implementation complexity when compared with its fully connected counterpart. Therefore, we mainly focus on the partially connected analog precoder at the relay node and the RF combiner at each SU. At the relay transmitter, it is assumed that Nr/NrRF antennas are connected to each of NrRF chains. Under this condition, FRF,2 becomes a block-diagonal matrix that contains a set of vectors p1, p2,…,pNrRF, each with length Nr/NrRF. Hence, the structure of FRF,2, under the partially connected architecture, is given as
(5)FRF,2=(1Nr)[p1⋯0⋮⋱⋮0⋯pNrRF], 
where pi∈ℂNr/NrRF×1 is the analog beamforming vector associated with the i-th sub-array. Owing to the relay-transmitted signal FRF,2FBB,2[n]yr1[n]∈ℂNr×1 to the SUs, the interference experienced by the PU can be represented as d2[n]=‖HPU[n]FRF,2FBB,2[n]yr1[n]‖F2. Using (4), the relay hybrid filter F[n]∈ℂNr×Nr is defined as F[n]=(FRF,2FBB,2[n])(FRF,1FBB,1[n])H, which can be compactly described as FRF,2FBB[n]FRF,1H, where FBB[n]FBB,2[n]FBB,1H[n]∈ℂNrRF×NrRF is the combined baseband processing matrix at the relay node. The received signal ydk+[n]∈ℂNdk×1 at the k-th SU under the assumption of block-fading channel can be characterized as
(6)ydk+[n]=Gdk[n]F[n](H[n]VRF∑k=1KVBB,k[n]sk[n]+nr[n])+zk[n], 
where Gdk[n]∈ℂNdk×Nr denotes the frequency domain channel between the relay node and the k-th SU, and zk[n]∈ℂNdk×1 specifies the ZMCSCG noise with variance σk2, i.e., zk[n]~CN(0, σk2INdk). The baseband equivalent signal ydk[n]∈ℂNs×1 after passing ydk+[n] in (6) through the hybrid combiner Wdk[n]=WRF,kWBB,k[n]∈ℂNdk×Ns at the k-th SU is given as ydk[n]=(WRF,kWBB,k[n])Hydk+[n], where WRF,k∈ℂNdk×NdkRF, WBB,k[n]∈ℂNdkRF×Ns are the common analog RF and the frequency-selective digital baseband combiners at the k-th SU. Under the assumption of the partially connected structure, the RF combiner WRF,k is constrained to be a block-diagonal matrix as follows:(7)WRF,k=(1Ndk)[q1⋯0⋮⋱⋮0⋯qNdkRF],
where qj∈ℂNdk/NdkRF×1 is the analog beamforming vector associated with the j-th sub-array. Here, Ndk/NdkRF antennas are connected to each of NdkRF chains at the k-th SU, and the block-diagonal matrix in (7) is composed of a set of vectors q1, q2,…,qNdkRF, each with length Ndk/NdkRF. Another useful representation of ydk[n] that facilitates writing the capacity expression is given as
(8)yck[n]={WBB,kH[n]Geq,k[n]FBB[n]Heq[n]∑k=1KVBB,k[n]sk[n]+WBB,kH[n]Geq,k[n]FBB[n]nr+[n]+WBB,kH[n]zk+[n], 
where Heq[n]=FRF,1HH[n]VRF∈ℂNrRF×NtRF denotes the equivalent baseband channel from the CRBS to relay station, Geq,k[n]=WRF,kHGdk[n]FRF,2∈ℂNdkRF×NrRF represents the equivalent baseband channel from the relay node to the k-th SU, nr+[n]=FRF,1Hnr[n]∈ℂNrRF×1 depicts the noise vector at the output of the relay RF combiner, and zk+[n]=WRF,kHzk[n]∈ℂNdkRF×1 specifies the noise vector at the k-th SU after passing through the analog RF processing matrix. It is worthwhile to mention that the noise distribution does not change after multiplication with the RF precoder/combiner, as given in [71]. This result leads to the conclusion that zk+[n] and nr+[n] follow the same distribution as that of zk[n] and nr[n], respectively. Using (8), the capacity expression of the k-th SU at the n-th sub-carrier can be represented as [70]
(9)Ck[n]=(12)log2det(INs+Xk[n]XkH[n](σr2Yk[n]YkH[n]+σk2WBB,kH[n]WBB,k[n])︸equivalent noise covariance matrix (=Rk[n])+ΞIUI[n]), 
where {Xk[n]=WBB,kH[n]Geq,k[n]FBB[n]Heq[n]VBB,k[n]Yk[n]=WBB,kH[n]Geq,k[n]FBB[n]nr+[n]Zk[n]=WBB,kH[n]Geq,k[n]FBB[n]Heq[n]∑j=1,j≠kKVBB,j[n]︸interuser interference (IUI) ,ΞIUI[n]=Zk[n]ZkH[n].

The pre-log factor (1/2) in (9) indicates that two time slots are required for signal transmission from the CRBS to K SUs. The sum spectral efficiency averaged over $N_{sub}$ sub-carriers can be described as
(10)$$R_{sum}=\left(\frac{1}{N_{sub}}\right)\overset{N_{sub}}{\underset{n=1}{\sum }}\left(\overset{K}{\underset{k=1}{\sum }}C_{k}\left[n\right]\right)=\left(\frac{1}{N_{sub}}\right)\overset{N_{sub}}{\underset{n=1}{\sum }}\left(\overset{K}{\underset{k=1}{\sum }}\overset{N_{s}}{\underset{i=1}{\sum }}\log_{2}\left\{{\left(1+{\left(SNIR\right)}_{k_{i}}\left[n\right]\right)}_{SD}\right\}\right),\;$$
where ${\left(SNIR\right)}_{k_{i}}\left[n\right]$ is the signal-to-noise plus interference ratio from the CRBS to the k-SU corresponding to the *i*-th data stream under the assumption of Gaussian signaling.

### 2.2. Channel Model

In this paper, the clustered channel model is considered to characterize the mm-wave channels [72]. The mm-wave channels exhibit sparse scattering or spatial selectivity due to high propagation losses. Additionally, deploying a large number of antennas in massive MIMO systems within a small physical region leads to high antenna correlation. Therefore, low-rank matrices can be used to characterize the sparse scattering nature of mm-wave channels. In sum, the traditional rich scattering Rayleigh fading channel is no longer applicable for modeling mm-wave channels. To capture the mathematical structure of the mm-wave propagation environment, the geometric channel based on the extended Saleh–Valenzuela model is adopted. Specifically, the mathematical formulation of the frequency domain channel matrix, considering the uniform planar array (UPA), for the n-th sub-carrier can be expressed as [51]
(11)$$H\left[n\right]=\beta \overset{N_{cl}-1}{\underset{i=0}{\sum }}\overset{N_{ray}}{\underset{l=1}{\sum }}\alpha_{il}a_{r}\left(\varphi_{il}^{r},\theta_{il}^{r}\right)\;\;a_{t}{\left(\varphi_{il}^{t},\theta_{il}^{t}\right)}^{H}\;e^{-j2\pi \frac{i.n}{N_{sub}}}\;,\;$$
where $\beta =\sqrt{\frac{N_{t}N_{r}}{N_{cl}N_{ray}}}$ describes the normalization factor, $N_{cl}$ and $N_{ray}$ illustrate the number of clusters and the number of rays in each cluster, *N_sub_* shows the total number of sub-carriers, $\alpha_{il}$~$CN\left(0,\sigma_{\alpha ,i}^{2}\right)$ specifies the complex gain, $\theta_{il}^{t}$ and $\theta_{il}^{r}$ denote the elevation angles of departure and arrival, and $\varphi_{il}^{t}$ and $\varphi_{il}^{r}$ represent the azimuth angles of departure and arrival of the l-th transmission path in the i-th propagation cluster. Moreover, $a_{t}\left(\varphi_{il}^{t},\theta_{il}^{t}\right)$ and $a_{r}\left(\varphi_{il}^{r},\theta_{il}^{r}\right)$ in (11) refer to the planar array response vectors at the transmitter and receiver, respectively. These array response vectors depend on the antenna array architecture, where each element needs to follow a constant modulus constraint. The uniform square planar array (USPA) with $\sqrt{N}\times \sqrt{N}$ antenna elements is considered in this work. Hence, the array response vector corresponding to the l-th ray in the i-th cluster is given as [51]
(12)$$a\left(\varphi_{il},\theta_{il}\right)=\left(\frac{1}{\sqrt{N}}\right){\left[1,\ldots ,e^{j\frac{2\pi }{\lambda }d\left(u\sin\varphi_{il}\sin\theta_{il}+v\cos\theta_{il}\right)},\ldots ,\;e^{j\frac{2\pi }{\lambda }d\left(\left(\sqrt{N}-1\right)\sin\varphi_{il}\sin\theta_{il}+\left(\sqrt{N}-1\right)\cos\theta_{il}\right)}\right]}^{T},$$
where λ and $d=\frac{\lambda }{2}$ are the signal wavelength and antenna spacing, and $0\leq u<\sqrt{N}$ and $0\leq v<\sqrt{N}$ are the antenna indices in the 2D plane. Note that the proposed algorithm and the obtained results can be extended to a uniform rectangular array (URA), and the corresponding array response vectors are given in [73].

### 2.3. Problem Formulation

The orientation of the proposed algorithm is to design a hybrid beamforming solution for mm-wave wideband relay-assisted MU-MIMO CRN. Furthermore, mixed and fully connected hybrid architectures are taken into consideration. It is important to mention that spectral efficiency and mean-squared error (MSE) are two optimization targets that are normally considered when the underlying problems involve the DF protocol at the relay station [55]. Note that the former is a crucial performance metric as far as hybrid transceiver designs are concerned. Hence, the prime objective of this work is to derive a set of hybrid processing components that maximize the sum spectral efficiency given in (10). It is also required to satisfy the transmit power constraint at the source while keeping interference experienced by the PU within a predefined threshold. Additionally, the analog RF beamforming components need to follow the element-wise constant amplitude constraints.

Therefore, the optimization problem can be formulated as follows:(13)$$\underset{{\left\{\begin{array}{c} V_{RF},V_{BB}\left[n\right],\;F_{RF,1},F_{BB,1}\left[n\right],\\ F_{BB,2}\left[n\right],F_{RF,2},\;W_{RF,k},W_{BB,k}\left[n\right] \end{array}\right\}}_{k=1,\;n=1}^{K,\;N_{sub}}}{\max}R_{sum}\;s.\;t.\;\left\{ \begin{array}{c} \left|V_{RF}(x,y)\right|=\frac{1}{\sqrt{N_{t}}},\left|W_{RF,k}(x,y)\right|=\frac{1}{\sqrt{N_{d_{k}}}},\\ \left|F_{RF,1}(x,y)\right|=\left|F_{RF,2}(x,y)\right|=\frac{1}{\sqrt{N_{r}}},\forall x,y,k,\\ F_{RF,2}=\left[\begin{array}{ccc} p_{1} & \cdots & 0\\ \vdots & \ddots & \vdots \\ 0 & \cdots & p_{N_{r}^{RF}} \end{array}\right],\\ W_{RF}=\left[\begin{array}{ccc} W_{RF,1} & \cdots & 0\\ \vdots & \ddots & \vdots \\ 0 & \cdots & W_{RF,K} \end{array}\right],\\ F_{BB,2}[n]=\left[\begin{array}{ccc} F_{BB,2}^{\left[1\right]}[n] & \cdots & 0\\ \vdots & \ddots & \vdots \\ 0 & \cdots & F_{BB,2}^{\left[K\right]}[n] \end{array}\right],\\ W_{BB}[n]=\left[\begin{array}{ccc} W_{BB,1}[n] & \cdots & 0\\ \vdots & \ddots & \vdots \\ 0 & \cdots & W_{BB,K}[n] \end{array}\right],\\ {\left\|V_{RF}V_{BB}[n]\right\|}_{F}^{2}\leq P_{s},\\ d_{1}[n]\leq J_{1},\\ d_{2}[n]\leq J_{2},\forall n \end{array},\right.$$
where $J_{1}$, $J_{2}$, and $P_{s}$ represent the interference threshold experienced by the PU due to the source-transmitted signal, the relay-transmitted signal, and the upper bound for the source-transmitted power, respectively. The solution of the optimization problem in (13) seeks a joint optimization over several complex matrix variables, namely $V_{RF}$, $V_{BB}\left[n\right]$, $F_{RF,1}$, $F_{BB,1}\left[n\right]$, $F_{BB,2}\left[n\right]$, $F_{RF,2}$, $W_{RF,k}$, and $W_{BB,k}\left[n\right]$ for each SU k and each sub-carrier n. It is worth mentioning that joint optimization, in this case, is usually NP-hard. Moreover, the element-wise constant modulus constraints imposed by the analog RF processing matrices, i.e., precoders and combiners, make the problem non-convex. Therefore, the global optimal solution is intractable. To address these challenges, an endeavor is made to divide the complicated optimization problem (13) into sub-problems. One sub-problem corresponds to the first transmission phase, i.e., from source to relay station, and deals with the optimization of complex variables $V_{RF}$, $V_{BB}\left[n\right]$, $F_{RF,1}$, and $F_{BB,1}\left[n\right]$. The other sub-problem focuses on deriving the decision variables $F_{BB,2}\left[n\right]$, $F_{RF,2}$, $W_{RF,k}$, and $W_{BB,k}\left[n\right]$ that correspond to the second transmission phase, i.e., from the relay to K SUs.

## 3. Hybrid Transceiver Design Based on SNR Maximization

The cooperative communication network, where the DF relay station is deployed between the source and destination, acts like a cascade of two sub-networks. Therefore, the part of the system from the source to the relay receiver corresponds to one sub-network, while the part of the system from the relay transmitter to the destination indicates the other sub-network. In the underlying hybrid transceiver, the decoded signal at the output of the relay hybrid combiner is applied to the input of the relay hybrid precoder. This process is referred to as cascading. Exploiting this property of the DF relay node, the section of the network from the CRBS to the SUs can be decomposed into two relatively independent sub-systems, which is the first step towards reducing the complexity of the original optimization problem (13). Let $R_{1}$ be the spectral efficiency of the point-to-point MIMO system from the source to the relay node and $R_{2}$ be the spectral efficiency of the point-to-multipoint MIMO system from the relay station to the SUs. It is possible to determine the overall sum rate of the two sub-systems by maximizing the minimum between $R_{1}$ and $R_{2}$. Taking advantage of this important feature, the sum rate of the entire SN can be expressed as [68]
(14)$$R_{sum}=\left(\frac{1}{2}\right)\min\left(R_{1},R_{2}\right).\;$$
Leveraging the separation of the sum rate into two parts, the optimization problem formulated in (13) can be divided into two sub-problems as
(15)$$\underset{{\left\{V_{RF},V_{BB}\left[n\right],F_{RF,1}F_{BB,1}\left[n\right]\right\}}_{n=1}^{N_{sub}}}{\max}R_{1}\;s.\;t.\;\{\begin{array}{c} Tr\left\{{\left(V_{RF}V_{BB}\left[n\right]\right)}^{H}\left(V_{RF}V_{BB}\left[n\right]\right)\right\}\leq P_{s},\;\forall n,\\ d_{1}\left[n\right]\leq J_{1},\\ \left|V_{RF}\left(x,y\right)\right|=\frac{1}{\sqrt{N_{t}}}\;,\left|F_{RF,1}\left(x,y\right)\right|=\frac{1}{\sqrt{N_{r}}},\;\forall \;x,y \end{array},\;$$
(16)$$\underset{{\left\{F_{BB,2}\left[n\right],F_{RF,2},\;W_{RF,k},W_{BB,k}\left[n\right]\right\}}_{k=1,\;n=1}^{K,\;\;N_{sub}}}{\max}R_{2}\;s.\;t.\;\left\{ \begin{array}{c} F_{RF,2}=\left[\begin{array}{ccc} p_{1} & \cdots & 0\\ \vdots & \ddots & \vdots \\ 0 & \cdots & p_{N_{r}^{RF}} \end{array}\right],\\ W_{RF}=\left[\begin{array}{ccc} W_{RF,1} & \cdots & 0\\ \vdots & \ddots & \vdots \\ 0 & \cdots & W_{RF,K} \end{array}\right],\\ F_{BB,2}[n]=\left[\begin{array}{ccc} F_{BB,2}^{\left[1\right]}[n] & \cdots & 0\\ \vdots & \ddots & \vdots \\ 0 & \cdots & F_{BB,2}^{\left[K\right]}[n] \end{array}\right],\\ W_{BB}[n]=\left[\begin{array}{ccc} W_{BB,1}[n] & \cdots & 0\\ \vdots & \ddots & \vdots \\ 0 & \cdots & W_{BB,K}[n] \end{array}\right],\\ d_{2}[n]\leq J_{2},\\ d_{2}[n]\leq J_{2},\\ \left|F_{RF,2}(x,y)\right|=\frac{1}{\sqrt{N_{r}}},\left|W_{RF,k}(x,y)\right|=\frac{1}{\sqrt{N_{d_{k}}}},\forall x,y. \end{array}\right.$$

### 3.1. Source Analog Precoder and Relay RF Combiner

The goal is to derive the RF precoder and combiner that maximize the sum spectral efficiency. Since the achievable rate $R_{1}$ is a function of complex matrix variables $\left(V_{RF},\;{\left\{V_{BB}\left[n\right]\right\}}_{n=1\;}^{\;N_{sub}\;},\;F_{RF,1},\;{\left\{F_{BB,1}\left[n\right]\right\}}_{n=1\;}^{\;N_{sub}\;}\right)$, joint optimization of the sum rate is computationally inefficient. Therefore, a decoupled approach is exploited to design the required analog beamforming components, which allows converting the sub-optimization problem (15) into the following form.
(17)$$\underset{{\left\{V_{RF},V_{BB}\left[n\right]\right\}}_{\;n=1}^{\;\;N_{sub}}}{\max}\left(\frac{1}{N_{sub}}\right)\overset{N_{sub}}{\underset{n=1}{\sum }}\log_{2}\det\left(I_{KN_{s}}+\frac{1}{\sigma_{r}^{2}}\left(H\left[n\right]V_{RF}V_{BB}\left[n\right]\right){\left(H\left[n\right]V_{RF}V_{BB}\left[n\right]\right)}^{H}\right)\;s.\;t.\;\{\begin{array}{c} Tr\left\{{\left(V_{RF}V_{BB}\left[n\right]\right)}^{H}\left(V_{RF}V_{BB}\left[n\right]\right)\right\}\leq P_{s},\;\forall n,\\ d_{1}\left[n\right]\leq J_{1},\\ \left|V_{RF}\left(x,y\right)\right|=\frac{1}{\sqrt{N_{t}}}\;,\left|F_{RF,1}\left(x,y\right)\right|=\frac{1}{\sqrt{N_{r}}},\;\forall \;x,y \end{array}.\;$$
According to Lemma 1 [41], the optimization problem in (17) can be approximated as
(18)max{VRF,VBB[n]} n=1  Nsub(1Nsub)∑n=1Nsub‖(VFDopt[n])HVRFVBB[n]‖F2s. t. {Tr{(VRFVBB[n])H(VRFVBB[n])}≤Ps, ∀n,d1[n]≤J1,|VRF(x,y)|=1Nt ,|FRF,1(x,y)|=1Nr, ∀ x,y. 
where VFDopt[n]∈ℂNt×KNs is the fully digital precoder corresponding to the n-th sub-carrier. If the hybrid beamformer VRFVBB[n] is sufficiently close to the full complexity precoder with considerably high SNR, then the optimization problem (18) becomes equivalent to (17). Furthermore, assuming VBB[n] as a unitary matrix allows us to transform the problem (18) into another equivalent form as
(19)maxVRF(1Nsub)∑n=1Nsub‖(VFDopt[n])HVRF‖F2s. t. {Tr{(VRFVBB[n])H(VRFVBB[n])}≤Ps, ∀n,d1[n]≤J1,|VRF(x,y)|=1Nt ,∀ x,y. 

Since the frequency-domain MIMO channels ${\left\{H\left[n\right]\right\}}_{\;n=1}^{\;\;N_{sub}}$ have the same row/column space [49], $V_{RF}$ is common to all sub-carriers. From this perspective, $V_{RF}$ can be considered as a representation of the mentioned column space, and this observation inspires us to develop the analog beamformer using the principal component analysis framework. Under this framework, the data set matrix $V_{data}=\left[V_{FD}^{opt}\left[1\right],\;V_{FD}^{opt}\left[2\right],\;\ldots ,V_{FD}^{opt}\left[N_{sub}\right]\;\right]$ can be employed to determine the principal components that constitute $V_{RF}$. Applying singular value decomposition (SVD) on $V_{data}$ leads to a stable solution with low complexity. Therefore, $V_{data}=U_{V_{data}}\Sigma_{V_{data}}V_{V_{data}}^{H}$ and the sub-optimal solution can be given as follows:(20)$$V_{RF}=\left(\frac{1}{\sqrt{N_{t}}}\right)\exp\left\{j\;arg\left(U_{V_{data}}\left(:,1:N_{t}^{RF}\right)\right)\right\}.$$

It is worth highlighting that the obtained RF beamforming vectors have the largest projections on the respective eigenmodes, i.e., they cast maximum energy along those eigenmode directions [74]. Additionally, they help achieve maximum beamforming gain and keep the interference that PU experiences within a predetermined threshold. We can also find a sub-optimal solution to the problem (19) by deriving the lower bound of the objective function. An alternative solution to design the $V_{RF}$ is as follows:(21)$$\overset{N_{sub}}{\underset{n=1}{\sum }}{\left\|{\left(V_{FD}^{opt}\left[n\right]\right)}^{H}V_{RF}\right\|}_{F}^{2}={\left\|{\left(V_{FD}^{opt}\left[1\right]\right)}^{H}V_{RF}\right\|}_{F}^{2}+\cdots +\|{\left(V_{FD}^{opt}\left[N_{sub}\right]\right)}^{H}V_{RF}{\|}_{F}^{2}\;\geq \left\|(\frac{{\left(V_{FD}^{opt}\left[1\right]\right)}^{H}+\ldots +{\left(V_{FD}^{opt}\left[N_{sub}\right]\right)}^{H}}{N_{sub}}\right)V_{RF}{\|}_{F}^{2}=\|V_{opt}V_{RF}{\|}_{F}^{2},$$
where $V_{opt}=\left(\frac{{\left(V_{FD}^{opt}\left[1\right]\right)}^{H}+\ldots +{\left(V_{FD}^{opt}\left[N_{sub}\right]\right)}^{H}}{N_{sub}}\right)$. The sub-optimal solution to the problem (19) can also be expressed as $V_{RF}=\left(\frac{1}{\sqrt{N_{t}}}\right)\exp\left\{j\;arg{\left(V_{opt}\right)}^{H}\right\}$. Based on the derived $V_{RF}$ and ${\left\{V_{BB}\left[n\right]\right\}}_{n=1}^{N_{sub}}$, the relay RF combiner $F_{RF,1}$ can be obtained by maximizing the received SNR. Using (3), the achievable rate can be expressed as
(22)$$R_{1}=\left(\frac{1}{N_{sub}}\right)\overset{N_{sub}}{\underset{n=1}{\sum }}\left(\log_{2}\det\left(I_{KN_{s}}+{R_{N_{1}}}^{-1}\left(F_{BB,1}^{H}\left[n\right]F_{RF,1}^{H}H\left[n\right]V_{RF}V_{BB}\left[n\right]\right){\left(F_{BB,1}^{H}\left[n\right]F_{RF,1}^{H}H\left[n\right]V_{RF}V_{BB}\left[n\right]\right)}^{H}\right)\right),$$
where $R_{N_{1}}=\left({\left(F_{RF,1}F_{BB,1}\left[n\right]\right)}^{H}\left(F_{RF,1}F_{BB,1}\left[n\right]\right)\sigma_{r}^{2}\right)$ is the noise covariance matrix. It is worth noting that the analog beamforming matrix does not change the noise distribution [71]. Additionally, assuming $F_{BB,1}\left[n\right],\;\forall n$ as a unitary matrix keeps the noise power the same, i.e., $\sigma_{r}^{2}$ after applying the hybrid combiner $\left(F_{RF,1}F_{BB,1}\left[n\right]\right)$ at the relay node. Hence, the total post-processing SNR, which is determined after hybrid combining, is given as
(23)$$SNR=\frac{\|F_{BB,1}^{H}\left[n\right]F_{RF,1}^{H}H\left[n\right]V_{RF}V_{BB}{\left[n\right]\|}_{F}^{2}}{\sigma_{r}^{2}}.\;$$
It is known that capacity expression in MIMO systems is directly related to SNR. Therefore, the sub-optimization problem (15) for deriving $F_{RF,1}$ can be approximately written as
(24)$$\underset{{\left\{F_{RF,1},F_{BB,1}\left[n\right],\right\}}_{n=1}^{N_{sub}}}{\max}\left(\frac{1}{N_{sub}}\right)\overset{N_{sub}}{\underset{n=1}{\sum }}\left(\frac{\|F_{BB,1}^{H}\left[n\right]F_{RF,1}^{H}H\left[n\right]V_{RF}V_{BB}{\left[n\right]\|}_{F}^{2}}{\sigma_{r}^{2}}\right)\;s.\;t.\;\{\begin{array}{c} Tr\left\{{\left(V_{RF}V_{BB}\left[n\right]\right)}^{H}\left(V_{RF}V_{BB}\left[n\right]\right)\right\}\leq P_{s},\;\forall n,\\ d_{1}\left[n\right]\leq J_{1},\\ \left|F_{RF,1}\left(x,y\right)\right|=\frac{1}{\sqrt{N_{r}}},\;\forall \;x,y \end{array}.\;$$
Furthermore, the sub-problem in (24) for maximizing the post-processing SNR can be equivalently transformed into the received power maximization problem as follows:(25)$$\underset{{\left\{F_{RF,1},F_{BB,1}\left[n\right],\right\}}_{n=1}^{N_{sub}}}{\max}\left(\frac{1}{N_{sub}}\right)\overset{N_{sub}}{\underset{n=1}{\sum }}Tr\left\{\left(F_{BB,1}^{H}\left[n\right]H_{eq}\left[n\right]V_{BB}\left[n\right]\right){\left(F_{BB,1}^{H}\left[n\right]H_{eq}\left[n\right]V_{BB}\left[n\right]\right)}^{H}\right\}\;s.\;t.\;\{\begin{array}{c} Tr\left\{{\left(V_{RF}V_{BB}\left[n\right]\right)}^{H}\left(V_{RF}V_{BB}\left[n\right]\right)\right\}\leq P_{s},\;\forall n,\\ d_{1}\left[n\right]\leq J_{1},\\ \left|F_{RF,1}\left(x,y\right)\right|=\frac{1}{\sqrt{N_{r}}},\;\forall \;x,y \end{array}.\;$$

Again, joint optimization of FRF,1 and {FBB,1[n]}n=1Nsub in (24) is challenging due to the coupling between these processing matrices. This gives motivation for designing these beamforming components separately. Therefore, the digital baseband processing components {VBB[n], FBB,1[n]}n=1Nsub in (24) can be ignored following a decoupled approach to reduce the complexity associated with the solution. Once the frequency-independent relay analog combiner is derived, the standard method can be used to find these baseband processing matrices to minimize interference among transmitted data streams. Therefore, the optimization problem (25) can be suppressed as
(26)maxFRF,1(1Nsub)∑n=1NsubTr{Heq[n]HeqH[n]}=Tr{(FRF,1HH[n]VRF)(FRF,1HH[n]VRF)H}s. t. {Tr{(VRFVBB[n])H(VRFVBB[n])}≤Ps, ∀n,d1[n]≤J1,|FRF,1(x,y)|=1Nr, ∀ x,y. 
For designing the FRF,1, the objective function in (26) can be further simplified as
(27)maxFRF,1(1Nsub)∑n=1NsubTr{FRF,1HHtmp[n]HtmpH[n]FRF,1}s. t. {Tr{(VRFVBB[n])H(VRFVBB[n])}≤Ps, ∀n,d1[n]≤J1, |FRF,1(x,y)|=1Nr, ∀ x,y, 
where Htmp[n]=H[n]VRF. The product Htmp[n]HtmpH[n] in the objective function of (27) can be replaced with the matrix given as
(28)A=∑n=1Nsub(Htmp[n]HtmpH[n]Nsub), 
where A∈ℂNr×Nr is the mean of the covariance of frequency domain channels Htmp[n] taken over Nsub sub-carriers. Using (28), the optimization problem in (27) can be written as
(29)maxFRF,1Tr{FRF,1HA FRF,1}s. t. {Tr{(VRFVBB[n])H(VRFVBB[n])}≤Ps, ∀n,d1[n]≤J1, |FRF,1(x,y)|=1Nr, ∀ x,y. 
The function Tr{FRF,1HA FRF,1} in (29) can also be expressed as
(30)Tr{FRF,1HA FRF,1}=∑m=1NrRF(fRF,1H)[m]A(fRF,1)[m], 
where (fRF,1)[m]∈ℂNr×1 is the m-th analog beamforming vector of FRF,1, and (fRF,1H)[m]A(fRF,1)[m] is the m-th diagonal element of FRF,1HA FRF,1∈ℂNrRF×NrRF. Using (30), the problem in (29) can be converted into the equivalent vector form as
(31)max{(fRF,1)[m]}m=1NrRF∑m=1NrRF(fRF,1H)[m]A(fRF,1)[m]s. t. {Tr{(VRFVBB[n])H(VRFVBB[n])}≤Ps, ∀n,d1[n]≤J1, |(fRF,1)[m](l)|=1Nr, ∀ l∈{1,…,Nr}, 
where (fRF,1)[m](l) depicts the l-th element of the m-th column in FRF,1. The primary objective while designing FRF,1, is to decompose the channels into parallel sub-channels to ensure effective multi-steam transmission. The constant modulus constraints on the RF processing component pose a main challenge. Relaxing the constant amplitude constraints can lead to the optimal solution corresponding to FRF,1 by performing eigenvalue decomposition (EVD) on A (28). The optimal combiner can be determined by selecting the NrRF eigenvectors that correspond to the NrRF maximum eigenvalues. This matrix makes it possible to receive signals along the eigenmodes of the channel. Note that the eigenvectors cannot be directly used to rotate the signals when the element-wise constant amplitude constraints are taken into consideration. In this situation, the projection on each eigenmode can be considered a criterion for designing the RF beamforming vectors. The best beamforming matrix should be the one whose vectors have the largest projections on the respective eigenmodes. For instance, the EVD of A in (28) is characterized as
(32)A=TΣTH=∑r=1LλrtrtrH, 
where T∈ℂNr×Nr is the unitary matrix, and Σ∈ℂNr×Nr is a diagonal matrix that comprises eigenvalues. Furthermore, tr is the r-th vector in T, which is associated with the eigenvalue λr. From the perspective of mm-wave transmission, L is the rank of the channel and describes the number of propagation paths. To determine the RF beamforming vector (fRF,1)[m] in (31), there is a need to maximize the projection of the m-th data stream on the m-th eigenmode, i.e., |tmH(fRF,1)[m]|. The maximal projection can be obtained by formulating the optimization problem that minimizes the mean squared error (MSE) between the unknown vector (fRF,1)[m] and the corresponding unconstrained optimal vector. Therefore,
(33)min(fRF,1)[m]Tr{(tmopt−(fRF,1)[m])H(tmopt−(fRF,1)[m])}=‖tmopt−(fRF,1)[m]‖22s. t. |(fRF,1)[m](l)|=1Nr, ∀ l∈{1,…,Nr}, m∈{1,…,NrRF}, 
where tmopt=T(:,m)∈ℂNr×1 is the optimal combining vector corresponding to the maximum eigenvalue in Σ∈ℂNr×Nr (32), and ‖.‖2 stands for the L2-norm of a vector. The objective function in (33), using the property of L2-norm and trace operator, can be written as
(34)‖tmopt−(fRF,1)[m]‖22=2−2 Tr(ℝ[(fRF,1)[m](tmopt)H]). 

It is obvious from (34) that the minimum value can be obtained when ${\left(f_{RF,1}\right)}^{\left[m\right]}$ has the same phase-values as $t_{m}^{opt}$. Under this condition, the projection of ${\left(f_{RF,1}\right)}^{\left[m\right]}$ on $t_{m}^{opt}$ becomes maximum, and this information can be exploited for designing the required phase-only combining vector as [75]
(35)$${\left(f_{RF,1}\right)}^{\left[m\right]}=\left(\frac{1}{\sqrt{N_{r}}}\right)\exp\left\{j\;arg\left(p_{m}^{opt}\right)\right\},\;$$
where arg(.) denotes the argument operator. Hence, the $F_{RF,1}$ is formulated as
(36)$$F_{RF,1}=\left[{\left(f_{RF,1}\right)}^{\left[1\right]},\ldots ,{\left(f_{RF,1}\right)}^{\left[N_{r}^{RF}\right]}\right].\;$$

### 3.2. Baseband Beamformer Optimization

In the previous sub-section, the source analog precoder and the relay analog combiner were designed to maximize RF beamforming gain by maximizing projections along the respective eigenmodes. After deriving the above-mentioned phase-only processing components, it is possible to define the equivalent channel observed from the baseband processing units in the CRBS and the relay node. By diagonalizing this equivalent channel $H_{eq}\left[n\right]$ (8), the frequency-selective digital baseband precoders and combiners at the source and the relay node can be obtained. Hence, $H_{eq}\left[n\right]=U_{R}\left[n\right]\Sigma_{SR}\left[n\right]V_{S}^{H}\left[n\right]$, $V_{BB}\left[n\right]=V_{S}\left[n\right]\left(:,1:KN_{s}\right)$, $F_{BB,1}\left[n\right]=U_{R}\left[n\right]\left(:,1:KN_{s}\right)$. These baseband processing components minimize interference among transmitted data streams from the source to the relay station indirectly.

### 3.3. Relay RF Precoder and RF Combiners at SUs

In the second time slot, the received signal $y_{d_{k}}\left[n\right]$ at the k-th SU in terms of the transmitted signal $y_{r_{1}}\left[n\right]$ (3) from the relay can be written as
(37)$$y_{d_{k}}\left[n\right]={\left(W_{RF,k}W_{BB,k}\left[n\right]\right)}^{H}\left\{G_{d_{k}}\left[n\right]F_{RF,2}F_{BB,2}\left[n\right]y_{r_{1}}\left[n\right]+z_{k}\left[n\right]\right\}.\;$$

The compact representation of (37) is given as
(38)$$y_{d_{k}}^{c}\left[n\right]=W_{d_{k}}^{H}\left[n\right]G_{d_{k}}\left[n\right]F_{2}\left[n\right]s^{R}\left[n\right]+W_{d_{k}}^{H}\left[n\right]z_{k}\left[n\right].\;$$

The received signal at the k-th SU user can also be represented as
(39)y¯dk[n]=WdkH[n]Gdk[n]F2k[n]skR[n]+∑j=1,j≠kKWdjH[n]Gdj[n]F2j[n]sjR[n]+WdkH[n]zk[n], 
where Wdk[n]=WRF,kWBB,k[n]∈ℂNdk×Ns, F2[n]=FRF,2FBB,2[n]∈ℂNr×KNs, F2k[n]=FRF,2FBB,2k[n], and sR[n]=yr1[n]∈ℂKNs×1. Assuming Gaussian signaling, the expression of R2, using (39), is obtained as
(40)R2=log2det(INs+(RN2k)−1(WBB,kH[n]Geq,k[n]FBB,2[k][n])(WBB,kH[n]Geq,k[n]FBB,2[k][n])H),
where RN2k=WdkH[n](∑j≠k(Gj[n]F2j[n])(Gj[n]F2j[n])H+σk2INdk)Wdk[n] shows the residual IUI plus noise. Using (40), the SNIR at the k-th SU after hybrid combining is expressed as
(41)(SNIR)k=Tr((WBB,kH[n]Geq,k[n]FBB,2[k][n])(WBB,kH[n]Geq,k[n]FBB,2[k][n])H)Tr(WdkH[n](∑j≠k(Gj[n]F2j[n])(Gj[n]F2j[n])H+σk2INdk)Wdk[n]). 

It is evident from (41) that direct optimization is a difficult task due to the dependence on complex matrix variables. Additionally, the RF beamforming solution at the relay node and each SU essentially need to follow the element-wise constant amplitude constraints. Therefore, an endeavor is made to approximate the expression in (41) to reduce the complexity of the optimization problem. After determining the RF processing components in (41), digital baseband precoding matrices $F_{BB,2}^{\left[k\right]}\left[n\right],\;k\in \left\{1,\ldots ,K\right\},$ can be obtained through the equivalent baseband channels $W_{RF,k}^{H}G_{d_{k}}\left[n\right]F_{RF,2}$, k∈{1,…,K}, using the block diagonalization (BD) technique. This implies that $\left(G_{k}\left[n\right]F_{RF,2}F_{BB,2}^{j}\left[n\right]\right){\left(G_{k}\left[n\right]F_{RF,2}F_{BB,2}^{j}\left[n\right]\right)}^{H}=0$, ∀j≠k. Therefore, the SNIR in (41) can be reduced as
(42)$${\left(\bar{SNIR}\right)}_{k}=\frac{Tr\left(\left(W_{BB,k}^{H}\left[n\right]G_{eq,k}\left[n\right]F_{BB,2}^{\left[k\right]}\left[n\right]\right){\left(W_{BB,k}^{H}\left[n\right]G_{eq,k}\left[n\right]F_{BB,2}^{\left[k\right]}\left[n\right]\right)}^{H}\right)}{Tr\left(\sigma_{k}^{2}W_{d_{k}}^{H}\left[n\right]W_{d_{k}}\left[n\right]\right)}.\;$$

It has been shown in [71] that RF processing components do not have any impact on the noise distribution, i.e., the noise vector shows the same variance after multiplication with the analog beam formers, as already mentioned. Moreover, the BD technique gives unitary matrices as the baseband processing components at the SUs. In conclusion, the noise power in (41) remains constant, and the optimization problem defined in (16) can be transformed into a received power maximization problem. Following this discussion and decoupled design methodology, the sub-problem in (16) can be approximately written as
(43)$$\underset{F_{RF,2},\;W_{RF,k}}{\max}\left(\frac{1}{N_{sub}}\right)\overset{N_{sub}}{\underset{n=1}{\sum }}\left(Tr\left\{\left(W_{RF,k}^{H}G_{d_{k}}\left[n\right]F_{RF,2}\right){\left(W_{RF,k}^{H}G_{d_{k}}\left[n\right]F_{RF,2}\right)}^{H}\right\}\right)\;s.\;t.\;\{\begin{array}{c} F_{RF,2}=\left[\begin{array}{ccc} p_{1} & \cdots & 0\\ \vdots & \ddots & \vdots \\ 0 & \cdots & p_{N_{r}^{RF}} \end{array}\right],W_{RF}=\left[\begin{array}{ccc} W_{RF,1} & \cdots & 0\\ \vdots & \ddots & \vdots \\ 0 & \cdots & W_{RF,K} \end{array}\right],\\ d_{2}\left[n\right]\leq J_{2},\\ \left|F_{RF,2}\left(x,y\right)\right|=\frac{1}{\sqrt{N_{r}}}\;,\left|W_{RF,k}\left(x,y\right)\right|=\frac{1}{\sqrt{N_{d_{k}}}},\;\forall \;x,y \end{array}.\;$$

When a large number of antennas are installed at the relay node, it will become highly probable that FRF,2FRF,2H≈NrIr [43]. Furthermore,
(44)Ck=(1Nsub)∑n=1NsubGdk[n]GdkH[n]∈ℂNdk×Ndk. 

Under the assumption of the large antenna array and the mean of the covariance of frequency-domain channels (44), the problem in (43) can be expressed as
(45)$$\underset{\;W_{RF,k}}{\max}Tr\left(W_{RF,k}^{H}C_{k}\;W_{RF,k}\right)\;s.\;t.\;\{\begin{array}{c} W_{RF}=\left[\begin{array}{ccc} W_{RF,1} & \cdots & 0\\ \vdots & \ddots & \vdots \\ 0 & \cdots & W_{RF,K} \end{array}\right],\\ d_{2}\left[n\right]\leq J_{2},\\ \left|W_{RF,k}\left(x,y\right)\right|=\frac{1}{\sqrt{N_{d_{k}}}},\;\forall \;x,y \end{array}.$$

Except for the diagonal structure of WRF,k, the problem in (45) is the same as that of (29), and hence, the same solution strategy can be applied to find the partially connected analog RF combiner at the k-th SU. To avoid repetition, the main design steps are shown here. The EVD of Ck in (44) is given as
(46)Ck=LΣ2LH,
where L∈ℂNdk×Ndk is the unitary matrix, and Σ2∈ℂNdk×Ndk is a diagonal matrix that contains eigenvalues. Let Lkopt∈ℂNdk×Ns be the optimal combiner at the k-th SU that corresponds to the maximum eigenvalue in Σ2, which is given as
(47)Lkopt=[l1,l2,…,lNs]. 

The required RF combiner $W_{RF,\;k},\;k\in \left\{1,\ldots ,K\right\}$ at the k-th SU can be determined by finding its projection as close as possible to the optimal combiner in (47). This target can be achieved by formulating the MMSE problem as
(48)$$\underset{W_{RF,\;k},\;k\in \left\{1,\ldots ,K\right\}}{\min}{\left\|\left[\begin{array}{ccc} L_{1}^{opt}-W_{RF,1} & \cdots & 0\\ \vdots & \ddots & \vdots \\ 0 & \cdots & L_{K}^{opt}-W_{RF,K} \end{array}\right]\right\|}_{F}^{2}=\underset{W_{RF,\;k}}{\min}\overset{K}{\underset{k=1}{\sum }}\|L_{k}^{opt}-W_{RF,k}{\|}_{F}^{2}\;=\underset{W_{RF,\;k}}{\min}\overset{K}{\underset{k=1}{\sum }}\left(Tr\left\{{\left(L_{k}^{opt}-W_{RF,k}\right)}^{H}\left(L_{k}^{opt}-W_{RF,k}\right)\right\}\right)\;s.\;t.\;\left|W_{RF,k}\left(x,y\right)\right|=\frac{1}{\sqrt{N_{d_{k}}}}.\;$$

The minimum of the cost function in (48) occurs when WRF,k has the same phase values as Lkopt. Therefore, WRF,k can be obtained using the following relation.
(49)WRF,k=(1Nt)exp{j arg(Lkopt)}. 
To adjust FRF,2 in accordance with WRF,k, ∀k, the optimization problem based on received power maximization is approximated as
(50)maxFRF,2(1Nsub)∑n=1Nsub(Tr{(WRFHG[n]FRF,2)H(WRFHG[n]FRF,2)})s. t. {FRF,2=[p1⋯0⋮⋱⋮0⋯pNrRF],d2[n]≤J2,|FRF,2(x,y)|=1Nr ,∀ x,y , 
where WRF=bd[WRF,1,…,WRF,K]∈ℂKNdk×KNdkRF is a block diagonal matrix and G[n]=[Gd1T[n],…,GdKT[n]]T∈ℂKNdk×Nr. Just like (45), this problem can also be transformed into the following form:(51)max FRF,2Tr(FRF,2HD FRF,2)s. t. {FRF,2=[p1⋯0⋮⋱⋮0⋯pNrRF],d2[n]≤J2,|FRF,2(x,y)|=1Nr ,∀ x,y,
where D∈ℂNr×Nr is defined as
(52)D=(1Nsub)∑n=1NsubGH[n]WRFWRFHG[n]. 
When EVD is performed on D, then
(53)D=EΣ3EH, 
where D∈ℂNr×Nr is the unitary matrix and Σ3∈ℂNr×Nr is a diagonal matrix that contains eigenvalues. Using (53), the unconstrained optimal precoder at the relay Fopt∈ℂNr×NrRF can be determined as Fopt=E(:,1:NrRF). But this optimal solution does not fulfill the essential condition of constant modulus constraints associated with the desired RF beamforming solution, i.e., FRF,2. However, it is possible to find the nearest point to the corresponding optimal precoder in high-dimensional space. This process helps in deriving the FRF,2 from the unconstrained precoding matrix Fopt. To achieve this design objective, a problem is defined for deriving the required analog beamformer at the relay node as follows:(54)minFRF,2‖Fopt−FRF,2‖F2=Tr{(Fopt−FRF,2)H(Fopt−FRF,2)}s. t. |FRF,2(x,y)|=1Nr ,∀ x,y. 
It is clear from (54) that the reconstruction loss cannot be fully avoided by achieving the exact lower bound, i.e., ‖Fopt−FRF,2‖F2=0, due to the element-wise non-convex constraints. Therefore, an endeavor is made to minimize the reconstruction error ‖Fopt−FRF,2‖F2≈0 in (54) to find FRF,2. The objective function in (54) can be written as
(55)Tr{(Fopt−FRF,2)H(Fopt−FRF,2)}=‖Fopt‖F2+‖FRF,2‖F2−Tr{2ℝ(FRF,2FoptH)}=‖Fopt‖F2+‖FRF,2‖F2−2∑x=1Nr∑y=1NrRFℝ{FRF,2(x,y)Fopt*(x,y)}. 
It is obvious from (55) that the minimum value of ‖Fopt−FRF,2‖F2 can only be obtained when each ℝ{FRF,2(x,y)Fopt*(x,y)} attains its maximum value. As Fopt(x,y)=|Fopt(x,y)|exp{jarg(Fopt(x,y))} and FRF,2(x,y)=|FRF,2(x,y)|exp{jarg(FRF,2(x,y))}, therefore,
(56)ℝ{FRF,2(x,y)Fopt*(x,y)}=|Fopt(x,y)||FRF,2(x,y)|cosψ≤|Fopt(x,y)||FRF,2(x,y)|, 
where ψ=arg(FRF,2(x,y))−arg(Fopt(x,y)). When arg(FRF,2(x,y))=arg(Fopt(x,y)) then cosψ=1, and this condition leads (56) to its maximum value. In summary, the required RF beamformer FRF,2 shares the element-wise phase of the corresponding optimal beamformer Fopt. Hence, FRF,2 is given as
(57)FRF,2=(1Nr)exp{jarg(Fopt)}. 

### 3.4. Digital Baseband Precoding and Combining

The section of the secondary network (SN) from the relay node to the SUs can be transformed in terms of the baseband equivalent channels Geq, k[n]=WRF,kHGk[n]FRF,2, ∀k,n, as FRF,2 and WRF,k have already been derived. Taking advantage of this transformation, the frequency-selective low-dimensional baseband processing matrices FBB,2[n] and WBB,k[n] can be designed using conventional techniques by exploiting Geq, k[n],∀k, n. Minimizing the impact of IUI is the primary objective behind the design of baseband processing components. This goal can be achieved by dividing the MU-MIMO channel into multiple SU-MIMO channels, and the block diagonalization (BD) technique leads to the required solution [76]. The overall channel GeqRD[n]∈ℂNrRF×KNdkRF from the relay to K SUs can be defined as GeqRD[n]=[Geq,1T[n],…,Geq,KT[n]]T∈ℂNrRF×KNdkRF. The constraint Geq,jRD[n]FBB,2[k][n]=0, j≠k needs to be satisfied to eliminate the IUI, where FBB,2[k][n] denotes the baseband precoder at the relay node corresponding to the k-th SU. This constraint demands that the received signal by the k-th user must fall in the null space of other user channels.

Define G˜eq,kRD[n]=[Geq,1T[n],…,Geq,k−1T[n],Geq,k+1T[n],…,Geq,KT[n]]T∈ℂNrRF×(K−1)NdkRF that contains the individual baseband equivalent channels except for the intended user. Performing SVD on G˜eq,kRD[n] leads to the desired null space as follows
(58)G˜eq,kRD[n]=(U˜eq,kRD[n])[Σ˜eq,kRD[n]000][(V˜eq,kRD[n])(1)(V˜eq,kRD[n])(0)]H, 
where (V˜eq,kRD[n])(1) represents the subspace orthogonal bases, and (V˜eq,kRD[n])(0) denotes the null space orthogonal bases of G˜eq,kRD[n]. Therefore,
(59)G˜eq,kRD[n](V˜eq,iRD[n])(0)={0, i≠kG˜eq,kRD[n](V˜eq,iRD[n])(0), i=k}. 
Applying SVD on Geq, k[n](V˜eq,kRD[n])(0) gives
(60)Geq, k[n](V˜eq,kRD[n])(0)=(Ueq,kRD[n])[Σeq,kRD[n]000][(Veq,kRD[n])(1)(Veq,kRD[n])(0)]H,∀k. To eliminate IUI, (Veq,kRD[n])(1) corresponding to the non-zero singular values is selected for designing the precoding matrix. Therefore, the FBB,2[k][n] is given as
(61)FBB,2[k][n]=(V˜eq,kRD[n])(0)(Veq,kRD[n])(1), ∀k. 

## 4. Complexity Analysis

Complexity falls into two main categories: (1) hardware implementation complexity and (2) computational complexity. From the perspective of hardware complexity, a fully digital array requires one dedicated RF chain per antenna. For instance, $N_{t}^{RF}=N_{t}$ RF chains are required for the digital implementation of the source node. It is evident from the above-mentioned condition that a hybrid architecture demands significantly less hardware complexity, as it considerably reduces the number of RF chains, i.e., $N_{t}^{RF}\ll N_{t}$. Among different hybrid structures, the fully connected one employs $N_{r}^{RF}N_{r}$ phase shifters, which indicates that each RF chain is connected to all antennas in an array. On the other hand, the partially connected structure uses $N_{r}$ phase shifters. This reduction in the number of phase shifters indicates the lower hardware implementation complexity of partially connected architectures. The proposed scheme employs a combination of fully connected and partially connected structures at the relay node, which is referred to as a mixed architecture. Additionally, the hybrid combiner of each SU also uses a partially connected structure. Therefore, the suggested method shows less hardware complexity when compared with its fully connected counterpart.

In the previous section, a hybrid broadband mm-wave beamforming design was proposed for mixed-structure and fully connected relay-assisted MU-MIMO networks. Also, the CR communication framework was taken into consideration for efficient spectrum utilization. The primary objective behind this technique was to avoid spectral congestion that might occur in the presence of a huge number of connected devices. In addition, the suggested architecture leads to a cost-effective, energy-efficient, and low-complexity solution.

A brief analysis of the computational complexity of the proposed scheme and a comparison with other existing hybrid beamforming techniques are provided in this section. The derivation of the common analog beamforming solution and frequency-dependent digital baseband processing component in wideband systems determines the overall computational complexity. The frequency-independent phase-only beamformer can be obtained by taking the average of the covariance matrices of frequency domain channels, and this operation requires matrix addition and multiplication. Considering matrix multiplication as a major contributing factor, the computational complexity for designing the RF processing components at different communicating nodes is in the order of matrix multiplication. Despite knowing that complex multiplication operations are computationally more expensive compared to complex addition operations, they are both considered one floating-point operation in evaluating the computational cost. For instance, the resultant matrix in (28) contains the product $H_{tmp}\left[n\right]H_{tmp}^{H}\left[n\right]$, which requires approximately $N_{r}^{2}N_{t}$ floating-point operations for its computation. Moreover, eigendecomposition needs to be performed to extract the optimal phase values to design the RF beamforming matrices at various communicating nodes. Hence, the derivation of the analog beamformer requires computational complexity in the order of $O\left\{\max\left({N_{r}}^{3},{N_{r}}^{2}N_{t}N_{sub}\right)\right\}$. Furthermore, the computational cost associated with the evaluation of the baseband processing component is in the order of $O\left({\left(N_{r}^{RF}\right)}^{3}\right)$. Therefore, the total complexity of the proposed approach can be approximated as follows:(62)$$C^{T}\approx N_{r}^{2}N_{t}+{N_{r}}^{2}N_{t}N_{sub}+N_{d_{k}}^{2}N_{r}+N_{d_{k}}^{2}N_{r}N_{sub}+{\left(N_{r}^{RF}\right)}^{3}+{\left(N_{d_{k}}^{RF}\right)}^{3}.\;$$

The computational cost of hybrid transceiver design for OFDM-based large-scale MIMO systems [43] can be expressed as
(63)$$N_{sub}\left(N_{s}^{3}+N_{t}^{2}N_{r}+N_{t}\left({N_{s}}^{2}+{N_{r}}^{2}\right)\right)+N_{s}\left({N_{t}}^{2}+{N_{r}}^{2}\right).$$

Finally, the hybrid beamforming technique proposed in [41] demands the following computational complexity:(64)$$N_{s}^{4}+N_{s}^{3}\left(N_{t}+N_{r}\right)+{\left(N_{t}N_{r}N_{s}\right)}^{2}+N_{sub}\left(N_{s}^{3}+N_{t}^{2}N_{r}+N_{r}^{2}N_{s}\right).$$

Table 3 shows the complexities of different hybrid precoding methods. On the other hand, Figure 2 illustrates the computational complexity of the proposed hybrid transceiver and the hybrid processing techniques developed in [41,43,47,50] as a function of the number of antennas. To obtain these curves, computer simulations are conducted by changing the number of antennas over a wide range while keeping these parameters ($N_{sub}=64$, $N_{t}^{RF}=N_{r}^{RF}=KN_{s}$, $N_{d}^{RF}=N_{s}=5$, $N_{d_{k}}=9$, and K=5) constant. It is clear from the obtained results that the proposed method shows less computational cost compared to the algorithms presented in [41,43,47,50]. There is a large performance gap between the proposed approach and the algorithm developed in [43]. Also, this gap increases with the number of antennas. For instance, the computational complexity of the proposed scheme is approximately 40 times less than the hybrid transceiver suggested in [43] when 100 antennas are deployed at the source and relay station. Similarly, the computational cost of the proposed technique is approximately 87 times less than the hybrid precoding design in [43] with 225 antennas. In addition, a small performance gap is observed when compared with the algorithm given in [47].

## 5. Computer Simulations

This section provides numerical results to illustrate the effectiveness of the proposed hybrid precoding algorithm for mixed structures and the corresponding fully connected architecture. Specifically, the sum spectral efficiency and energy efficiency performance of the presented technique are compared with full-complexity digital precoding and other well-known hybrid beamforming designs. It is worth mentioning that computer simulations are conducted by changing system configuration parameters over a wide range, considering perfect CSI and imperfect CSI with different accuracy factors to show robustness. To obtain simulation results, the mm-wave MIMO channel is generated according to the model (12) given in Section 2. The number of propagation paths in each cluster and the number of clusters are set to $N_{ray}=10$ and $N_{c}=5$, respectively. The angles of departure and the angles of arrival in (12) are generated by following the Laplacian distribution with a uniformly distributed mean over [0, 2π). Moreover, the complex gain of each propagation path is assumed to satisfy the distribution $\alpha_{il}$~$CN\left(0,\sigma_{\alpha ,i}^{2}\right)$, where $\sigma_{\alpha ,i}^{2}=1\;\forall i$ is the average power of each cluster. In addition, the operating central frequency $f_{c}$ and the total number of sub-carriers $N_{sub}$ are set as follows: 28 GHz and 64, respectively. Note that the proposed algorithm is also applicable to any arbitrary number of sub-carriers (e.g., 128, 256, etc.). The upper bound for the transmit power of CRBS, transmission range, and normalized beamforming gain are given as $N_{s}$, 200–450 m, 1, respectively. All simulation results are obtained by averaging over 500 random channel realizations. It is worth highlighting that the estimated channel matrix $\bar{H}\left[n\right]$ can be characterized as [77]
(65)$$\bar{H}\left[n\right]=\delta \;H\left[n\right]+\sqrt{1-\delta^{2}}E,$$
where H[n] denotes the actual channel matrix, δ∈[0, 1] shows the accuracy factor of $\bar{H}\left[n\right]$, and E indicates the error matrix whose elements are independent and identically distributed (i.i.d) Gaussian random variables, i.e., CN(0, 1). The main simulation parameters are shown in Table 4.

### 5.1. Spectral Efficiency Evaluation

Figure 3 illustrates the sum spectral efficiency performance of the proposed hybrid beamforming as a function of SNR, considering mixed structure and fully connected architecture. The number of antennas deployed at the source and relay node is set as $N_{t}=N_{r}=256$, while 64 antennas are installed at each SU, i.e., $N_{d_{k}}=64$. It is assumed that the CRBS is serving K=4 SUs. Moreover, the number of transmitted data streams is set to $N_{s}=\left\{2,\;4,\;8\right\}$ under the assumption that $N_{t}^{RF}=N_{r}^{RF}=KN_{s}$ and $N_{d_{k}}^{RF}=N_{s}$, i.e., the number of RF chains is equal to the number of transmitted data streams at the respective communicating nodes. The obtained results demonstrate that the proposed technique achieves performance close to its fully digital counterpart with the fully connected architecture. On the other hand, the proposed scheme shows relatively lower sum spectral efficiency with the mixed structure when compared with the corresponding fully connected and full-complexity digital ones. It can also be seen from the obtained curves that performance degradation occurs in a gradual manner as channel estimation error increases, i.e., δ=0.9 (10%), 0.8 (20%), 0.7 (30%). Just like the mixed structure, the performance of the fully connected architecture follows the same pattern when channel estimation error increases in discrete steps. It is evident from the obtained curves that the performance gap increases slightly by increasing the number of transmitted data streams. However, the nearly consistent performance of the proposed algorithm, even with a large number of data streams per user, indicates that IUI is effectively suppressed.

Figure 4 plots the sum spectral efficiency of the proposed approach when 144 antennas are deployed at the source and relay nodes and 36 antennas are installed at each SU, i.e., $N_{t}=N_{r}=144$ and $N_{d_{k}}=36$. To generate simulation results, the least number of RF chains is considered at the respective communicating units, i.e., $N_{t}^{RF}=N_{r}^{RF}=KN_{s}$ and $N_{d_{k}}^{RF}=N_{s}$. The system is assumed to serve K=3 SUs, and the number of transmitted data streams to each user is set as *N_s_* = {2, 3, 4}.

It is clear from the obtained curves that the proposed hybrid transceiver achieves performance close to full-complexity beamforming with the fully connected architecture. However, a small degradation in performance occurs when the mixed structure is taken into consideration. It is also obvious from the obtained results that the proposed algorithm shows consistent performance regardless of changing system parameters. To describe the impact of channel estimation error on the performance of mixed hybrid structures, numerical results are obtained under imperfect CSI at different accuracy levels, i.e., δ=0.85, 0.75. Again, the decreasing accuracy factor leads to gradual degradation in performance, as shown in Figure 4.

Figure 5 describes the spectral efficiency performance of the proposed scheme by further increasing the number of users compared to the previous two cases. To obtain numerical results, the number of antennas at different communicating nodes and the number of transmitted data streams to each SU are set as follows: $N_{t}=N_{r}=144,\;N_{d_{k}}=16$ and $N_{s}=\left\{2,\;4\right\}$, respectively. It is assumed that the CRBS is serving six users, i.e., K=6. The proposed algorithm with the fully connected structure achieves performance close to the upper bound defined by fully digital precoding, while a small degradation in performance occurs with the mixed architecture. Furthermore, a gradual decrease in performance is observed when the accuracy of the estimated channel decreases by considering δ=0.9, 0.8, 0.7. It is obvious from Figure 5 that the obtained results follow a similar pattern as observed in the previous cases. This consistency in performance under changing system parameters indicates the usefulness of the suggested method.

Figure 6 depicts the achievable rate of the proposed approach, considering URA at communicating nodes. To generate numerical results, the number of antennas, transmitted data streams, and SUs are given as $N_{t}=N_{r}=200$, $N_{d_{k}}=50$, $N_{s}=5$, K=4, respectively. The obtained results demonstrate that the fully connected hybrid transceiver achieves performance close to the unconstrained fully digital precoding. Moreover, there is a minor performance gap between the mixed structure and the fully connected one, assuming perfect CSI. However, the performance of the proposed scheme decreases gradually by reducing the accuracy of the estimated channel matrix.

### 5.2. Impact of Number of Users on Spectral Efficiency

Figure 7 plots the sum spectral efficiency of the proposed hybrid transceiver as a function of the number of users in the secondary network (SN). The SNR value is set to 5 [dB], and the number of users increases from 2 to 16. Furthermore, the number of antennas at communicating nodes and the number of transmitted data streams to each SU are given as $N_{t}=N_{r}=256$, $N_{d_{k}}=16$, and $N_{s}=2$.

Since the effect of IUI increases with the number of users, the rate of change of sum spectral efficiency decreases accordingly. In the case of perfect CSI, the proposed hybrid precoding with the fully connected structure achieves performance close to fully digital beamforming. However, performance degradation occurs when the mixed architecture is taken into consideration. Additionally, the performance of the proposed method also decreases gradually when channel estimation error increases. Finally, the proposed algorithm outperforms hybrid precoding designs presented in [51,52].

### 5.3. Impact of Number of Antennas on Spectral Efficiency

Figure 8 shows the spectral efficiency performance of the proposed method by varying the number of antennas at the source, relay node, and SUs simultaneously. To visualize the impact of this change, the other system parameters, such as the number of users in the SN and transmitted data streams, are kept constant. It is known that high beamforming gain can be obtained by deploying a large number of transmit and receive antennas, as it results in a narrow beam formation. Numerical results are obtained by following an ascending order for the deployment of antennas at communicating units. The number of antennas at the source, relay station, and each SU is set as 64 (16) and 256 (64) to evaluate the sum spectral efficiency performance. While increasing the number of antennas at the source and relay, it is further assumed, without loss of generality, that $N_{t}=N_{r}$. Moreover, $N_{s}$ and K are set as 2 and 4, respectively. The spectral efficiency increases in a gradual fashion by increasing the number of antennas, irrespective of hybrid beamforming structures. It is obvious from the obtained results that the suggested fully connected hybrid precoding achieves performance close to full-complexity digital beamforming. On the other hand, the presented mixed hybrid processing structure achieves relatively lower performance when compared with its fully connected hybrid and conventional fully digital counterparts. Simulation results demonstrate that minor performance degradation occurs when the estimated channel matrix (82) with δ=0.9, 0.8, 0.7 is taken into consideration.

Figure 9 illustrates the sum spectral efficiency of the proposed approach as a function of the number of antennas at the source and relay node, while the number of antennas at the SUs is kept constant. The number of antennas installed at the source and the relay node is set to $N_{t}=N_{r}=\left\{64,\;144,\;256\right\}$. The other parameters like $N_{d_{k}}=16$, K=4, and $N_{s}=4$ are also kept constant while conducting computer simulations. Just like in the previous cases, the least number of RF chains is employed at the respective communicating nodes, i.e., $N_{t}^{RF}=N_{r}^{RF}=KN_{s}$ and $N_{d_{k}}^{RF}=N_{s}$, to obtain simulation results.

Under this condition, the system utilizes minimum power for signal transmission and reduces the cost factor as well. From the obtained curves in Figure 9, it is obvious that sum spectral efficiency increases with an increase in the number of antennas, but the rate of change decreases in a gradual manner. This behavior shows that saturation will occur at a significantly large number of antennas, where the rate of change of sum spectral efficiency becomes negligibly small. Simulation results indicate that the suggested fully connected transceiver achieves near-optimal performance, while relatively lower performance is obtained using its mixed-structure counterpart. The performance gap of the proposed method increases gradually by increasing the channel estimation error, irrespective of the hybrid precoding architecture, as shown in Figure 9.

The sum spectral efficiency of the proposed technique is plotted in Figure 10 as a function of the number of antennas at the SUs, while the number of data streams per SU, the number of SUs, the number of antennas at the relay station, and the number of antennas at the source are kept constant. To generate simulation results for visualizing the effect of this change, the fixed parameters are set as $N_{t}=N_{r}=256$, K=4, and $N_{s}=4$. Furthermore, each SU is equipped with the number of antennas $N_{d_{k}}=\left\{4,\;16,\;36,\;64\right\}$. The obtained performance curves show that spectral efficiency increases and the rate of change decreases by increasing the number of antennas at SUs. It is also evident from the obtained results that large-scale antenna arrays make it possible to enhance beamforming gain without deploying additional expensive RF chains. Like the previous results, the proposed algorithm with the fully connected structure achieves performance close to that of the full-complexity solution. Moreover, the proposed approach with the mixed structure obtains relatively lower performance at a small number of antennas in comparison to both fully digital and fully connected hybrid solutions. But it approaches the performance of the fully connected architecture with a relatively large number of antennas. In addition, the performance degradation occurs in a gradual fashion by increasing the channel estimation error, i.e., δ=0.9, 0.8, 0.7.

### 5.4. Impact of Number of Data Streams on Spectral Efficiency

Figure 11 depicts the sum spectral efficiency performance of the proposed approach by changing the number of data streams per user. In this case, the number of antennas at communicating units and the number of SUs are kept constant, and these parameters are set as $N_{t}=N_{r}=256$, $N_{d_{k}}=64$ and K=4. The number of data streams varies from 2 to 16, and performance curves are obtained at SNR = −10, 0, and 15 [dB]. Simulation results demonstrate that sum spectral efficiency increases and the rate of change decreases when the number of data streams per SU increases. In addition, there is a slight increase in the performance gap between the proposed hybrid precoding and the upper bound defined by full-complexity digital beamforming with an increase in the number of data streams. It is evident from the obtained results that the performance of the fully connected hybrid transceiver is comparatively better than the performance of the mixed hybrid structure. Also, the sum spectral efficiency of the suggested fully connected structure is approximately consistent at different SNR values. There is a minor decrease in the performance of the mixed hybrid structure at δ=0.8, 0.7 when compared with the corresponding curve using perfect CSI, which is obtained by taking the fully connected architecture into account.

### 5.5. Performance Evaluation with Other Hybrid Beamforming Algorithms

The sum spectral efficiency performance of the suggested method is compared with conventional fully digital beamforming and hybrid precoding techniques given in [51,52], as illustrated in Figure 12 and Figure 13. To show the usefulness of the presented technique, numerical results are obtained by changing the number of antennas at communicating nodes, the number of users in the secondary network, and the number of transmitted data streams to each user. Using the system parameters $N_{t}=N_{r}=144$, $N_{d_{k}}=36$, K=3, $N_{s}=2$, and $N_{sub}=64$, the sum spectral efficiency of various algorithms is plotted in Figure 12.

It is obvious from the performance evaluation curves that the proposed hybrid precoding with the fully connected structure can achieve spectral efficiency close to full-complexity digital beamforming. In addition, the proposed scheme shows relatively lower performance when the mixed hybrid structure is taken into account. It is obvious from Figure 12 that minor degradation in the performance of mixed hybrid precoding occurs, considering imperfect CSI at δ=0.85, 0.75. Furthermore, the proposed algorithm shows significantly better performance when compared with hybrid beamforming techniques suggested in [51,52].

Similarly, the sum spectral efficiency of different algorithms is plotted in Figure 13, where system parameters are set to $N_{t}=N_{r}=256$, $N_{d_{k}}=64$, K=4, $N_{s}=4$, and $N_{sub}=64$. It is evident from the obtained curves that the suggested hybrid precoding solution gives performance close to the upper bound when the fully connected structure is taken into consideration. On the other hand, degradation in performance occurs in the case of mixed hybrid architecture, as the underlying structure limits the flexibility of large-scale antenna systems. Additionally, a slight decrease in performance is observed owing to channel estimation error. Also, the proposed technique outperforms well-known hybrid beamforming designs presented in [51,52].

### 5.6. Energy Efficiency Evaluation

Energy consumption is of great practical importance for hybrid transceiver design in mm-wave relay-assisted MU-MIMO systems. This design aspect leads to sub-connected or mixed-structure hybrid beamforming, which enhances the energy efficiency (EE) in the domain of hybrid precoding when compared to the corresponding fully connected structure and full-complexity digital design. The number of phase shifters for a given number of RF chains, antennas, and data streams distinguishes both hybrid processing architectures. It is worthwhile to mention that the fully connected structure does better than the partially or mixed connected architecture in terms of spectral efficiency because it has more degrees of freedom (DoF) in the RF domain, as illustrated in the previous results. The number of phase shifters $N_{PS}$ for the relay node with fully connected and mixed structures can be expressed as
(66)$$N_{PS}=\{\begin{array}{c} N_{r}N_{r}^{RF},\;fully-connected\\ N_{r},\;partially-connected\; \end{array}.\;$$

The relay hybrid precoder and combiner use partially connected and fully connected structures, respectively. Similarly, the source hybrid beamformer and the hybrid combiner at each destination utilize the fully connected and sub-connected architectures, respectively. Hence, the overall network and relay fall into the category of mixed hybrid structures. It is obvious from (66) that the mixed-structure hybrid relay filter requires fewer phase shifters in comparison to its fully connected counterpart. This is an indication that the relay with the mixed architecture consumes less energy than the corresponding fully connected one. The EE is defined as [51]
(67)$$EE=\frac{R_{sum}}{P_{total}}\;\left(\mathrm{bps}/\mathrm{Hz}/W\right).\;$$

It is worth mentioning that the power amplifier (PA) and the low-noise amplifier (LNA) connected to each antenna on the transmitter and receiver, respectively, the RF chains and phase shifters on both transmitter and receiver, analog-to-digital converters (ADC) and digital-to-analog converters (DAC), and the digital baseband processor are the components in the relay hybrid filter that consume power. Therefore, the total consumed power of the DF relay station is given as
(68)$$P_{total}=2N_{r}^{RF}P_{RF}+N_{r}^{RF}\left(P_{ADC}+P_{DAC}\right)+N_{r}\left(P_{PA}+P_{LNA}\right)+2N_{PS}P_{PS}+2P_{BB},\;$$
where $P_{RF}$, $P_{ADC}$, $P_{DAC}$, $P_{PA}$, $P_{LNA}$, $P_{PS}$, and $P_{BB}$ show the power consumed by the RF chain, analog-to-digital converter, digital-to-analog converter, power amplifier, low-noise amplifier, phase shifter, and digital baseband processor, respectively. Specifically, the energy consumption of fully digital precoding $P_{FD}$, the proposed fully connected hybrid architecture $P_{FC}$, and the proposed mixed-structure hybrid transceiver $P_{PC}$ are shown in Table 5.

Using (67), the EE performance of different precoding architectures is compared numerically in this sub-section. This task can be accomplished by plotting the EE of the proposed hybrid beamforming structures and full-complexity precoding as a function of the number of RF chains and the number of antennas at the relay station. To evaluate sum spectral efficiency, the system parameters are set as $N_{t}=N_{r}=100$, $N_{d_{k}}=25$, K=4, $N_{s}=1$, SNR=10 [dB], and $N_{sub}=64$. Furthermore, the power simulation parameters are given as follows: $P_{BB}=200\;\mathrm{mW}$, $P_{RF}=100\;\mathrm{mW}$, $P_{LNA}=P_{PA}=100\;\mathrm{mW}$, $P_{PS}=10\;\mathrm{mW}$, $P_{DAC}=110\;\mathrm{mW}$, and $P_{ADC}=200\;\mathrm{mW}$. The obtained curves in Figure 14 indicate that the EE decreases as the number of RF chains increases. Owing to the high energy consumption, the EE of the DF-based fully connected architecture decreases more rapidly in comparison to the mixed relay hybrid structure. Also, the EE of the mixed structure decreases slowly as the number of phase shifters in the partially connected hybrid beamformer at the relay transmitter is equal to the number of antennas (66). Finally, the energy consumption is independent of the number of relay RF chains in fully digital beamforming, which always keeps its EE stable. Note that these results are consistent with the theoretical analysis.

Figure 15 demonstrates the EE comparison of different algorithms by changing the number of antennas at the relay station. To evaluate the sum rate, the system parameters are set as follows: $N_{t}=N_{r}=\left\{100,\;144,\;196,\;256\right\}$, $N_{d_{k}}=16$, K=4, $N_{s}=1$, SNR=10 [dB], and $N_{sub}=64$. It is clear from the obtained curves that the EE of the mixed DF-based scheme degrades more slowly than that of the corresponding fully connected hybrid and fully digital counterparts. Hence, the proposed mixed-structure technique achieves significantly higher EE by increasing the number of antennas.

Figure 16 and Figure 17 describe the relation between sum spectral efficiency and EE by varying the number of transmit antennas at the relay node. Furthermore, the number of data streams and the number of SUs are also changed to gain more insight into the above-mentioned relationship. Figure 16 plots the sum spectral efficiency and the corresponding EE when the number of antennas at the source and relay node are set as $N_{t}=N_{r}=\left\{64,100,\;144,\;196,\;256\right\}$, while $N_{d_{k}}=16$, K=4, $N_{s}=1$, SNR=10 [dB], and $N_{sub}=64$ are kept constant, whereas the number of antennas at the CRBS and the relay station is set as $N_{t}=N_{r}=\left\{64,\;144,\;256\right\}$, and the other constant parameters are given as $N_{d_{k}}=16$, K=8, $N_{s}=2$, SNR=10 [dB], and $N_{sub}=64$ to evaluate the sum spectral efficiency and the corresponding EE, as shown in Figure 17. It is evident from the obtained curves that an inverse relationship exists between spectral efficiency and EE. Again, these results are compatible with the theoretical analysis.

In the case of unconstrained fully digital precoding, the spectral efficiency is maximum, but this technique achieves minimum EE. The suggested mixed-structure hybrid transceiver, on the other hand, has the highest EE, but its spectral efficiency is lower than that of other beamforming methods.

Figure 18 illustrates the EE performance of different beamforming methods as a function of SNR. The simulation results are obtained by setting $N_{t}=N_{r}=144$, $N_{d_{k}}=36$, K=4, $N_{s}=2$, and $N_{sub}=64$. As expected, the presented mixed-structure hybrid beamforming technique shows the highest EE performance in comparison to the other two precoding methods.

### 5.7. Performance Evaluation at Higher Frequencies (Beam Splitting Effect)

The common (frequency-independent) analog RF processing components are required under wideband assumptions, as already mentioned. However, this design does not hold true when large antenna arrays are deployed, and signal transmission uses significantly higher frequencies. Since the array response vectors become frequency-dependent, the assumption of frequency-flat array response vectors does not remain valid at considerably higher frequencies. These conditions cause a beam-squint effect that splits beams into different physical directions at different frequencies. This process is also termed a spatial wideband effect in the time domain. Serious performance degradation may occur if this problem is not addressed properly while designing hybrid wideband transceivers. To describe the spatial wideband effect on the performance of the proposed algorithm, the following mm-wave channel matrix in the frequency domain is considered [1].
(69)$$H_{BS}\left[n\right]=\gamma \overset{N_{cl}}{\underset{i=1}{\sum }}\overset{N_{ray}}{\underset{l=1}{\sum }}\alpha_{il}a_{r}\left(\varphi_{i,l,n}\right)\;\;a_{t}{\left(\psi_{i,l,n}\right)}^{H}\;e^{-j2\pi \tau_{i,l}f_{n}}\;,\;$$
where $a_{t}\left(.\right)$ and $a_{r}\left(.\right)$ are the beam steering vectors at the source and destination, respectively. Notation *ψ*_*i*,*l*,*n*_ and $\varphi_{i,l,n}$ specify the angle of departure and arrival, respectively, corresponding to the l-th relay in the i-th cluster at sub-carrier frequency n. $\alpha_{il}$ and $\tau_{i,l}$ represent the path gain and path delay of the l-th propagation path in the i-th cluster, respectively, and $\gamma =\sqrt{\frac{N_{t}N_{r}}{N_{cl}N_{ray}}}$ stands for the normalization factor. The beam steering vectors $\;a_{t}\left(\psi_{i,l,n}\right)$ and $a_{r}\left(\varphi_{i,l,n}\right)$ can be characterized as [1]
(70)$$\;a_{t}\left(\psi_{i,l,n}\right)=\frac{1}{\sqrt{N_{t}}}{\left[1,\;e^{j2\pi \psi_{i,l,n}},\ldots ,e^{j\left(N_{t}-1\right)2\pi \psi_{i,l,n}}\right]}^{T},\;$$
(71)$$a_{r}\left(\varphi_{i,l,n}\right)=\frac{1}{\sqrt{N_{r}}}{\left[1,\;e^{j2\pi \varphi_{i,l,n}},\ldots ,e^{j\left(N_{r}-1\right)2\pi \varphi_{i,l,n}}\right]}^{T},\;$$
where $\psi_{i,l,n}$ and $\varphi_{i,l,n}$ can be determined using the closed-form expressions as follows:(72)$$\psi_{i,l,n}=\frac{f_{n}}{c}d\sin\left(\omega_{i,\;l}\right),\;$$
(73)$$\varphi_{i,l,n}=\frac{f_{n}}{c}d\sin\left(\mu_{i,\;l}\right),\;$$
where $d=\frac{c}{2f_{c}}=\frac{\lambda }{2}$ is the antenna spacing; λ is the wavelength at the central frequency $f_{c}$, which cannot be changed after the antenna fabrication; and c represents the speed of light. $\omega_{i,\;l}\in \left[0,\;2\pi \right)$ and $\mu_{i,\;l}\in \left[0,\;2\pi \right)$ denote the angle of departure and the angle of arrival at the source and destination, respectively, and $f_{n}=f_{c}+\frac{f_{s}}{N_{sub}}\left(n-1-\frac{N_{sub}-1}{2}\right)$, where $f_{s}$ shows the bandwidth. The simulation parameters for describing the spatial wideband effect on the performance of the proposed technique are given in Table 6.

Figure 19 demonstrates the impact of the spatial wideband effect on the sum spectral efficiency of the proposed technique, where the number of antennas $N_{t}=N_{r}=36$, $N_{d_{k}}=9$ and the number of transmitted data streams $N_{s}=3$ are taken into consideration. It is assumed that the system is serving four SUs, i.e., K=4. The optimal unconstrained fully digital beamforming defines the upper bound for comparing the performance of the presented approach. Numerical results illustrate that the proposed algorithm can achieve more than 91% of the optimal sum rate with a fully connected structure. Also, the suggested mixed hybrid structure can obtain approximately 86% of the optimal spectral efficiency.

Figure 20 compares the performance of the proposed algorithm with the corresponding full-complexity precoding by increasing the number of antennas at communicating nodes. The number of antennas, the number of SUs, and the number of data streams are given as $N_{t}=N_{r}=64$, $N_{d_{k}}=16$, K=4, and $N_{s}=4$, respectively. In this case, the fully connected structure shows more than 85% of the optimal sum rate, while the mixed structure achieves nearly 79% of the optimal spectral efficiency. Computer simulations demonstrate that performance decreases by increasing the number of antennas compared to the previous results, which is consistent with the theoretical analysis. In conclusion, the proposed scheme shows relatively lower performance under spatial wideband effects.

### 5.8. Impact of Uniform Circular Array (UCA)

Before describing the impact of the uniform circular array (UCA) on the performance of the proposed method, it is worth mentioning that wideband communication with UCA leads to a beam defocus effect, in contrast to the spatial wideband effect [78]. This results in a considerable performance loss if measures are not taken to tackle this challenge. To generate simulation results, the frequency domain wireless channel based on the Saleh–Valenzuela model [78] is given as follows:(74)$$H_{UCA}\left[m\right]=\eta \overset{L}{\underset{l=1}{\sum }}g_{l}a_{t}^{m}\left(\psi_{l}\right)\;a_{r}^{m}{\left(\varphi_{l}\right)}^{H}\;e^{-j2\pi \tau_{l}f_{m}},$$
where $a_{t}^{m}\left(.\right)$ and $a_{t}^{m}\left(.\right)$ are the array response vectors at the source and destination, respectively. The complex gain and delay of the l-th propagation path are denoted by $g_{l}$ and $\tau_{l}$, respectively, while $\eta =\sqrt{N/L}$ stands for the normalization factor, where N shows the number of antenna elements in UCA. The angle of departure and the angle of arrival corresponding to the l-th path is represented as $\psi_{l}$ and $\varphi_{l}$, respectively. The beam steering vector considering the physical direction ψ can be expressed as follows:(75)$$a_{t}^{m}\left(\psi \right)=\frac{1}{\sqrt{N}}{\left[e^{j\rho_{m}\cos\left(\psi -\Phi_{0}\right)},\ldots ,e^{j\rho_{m}\cos\left(\psi -\Phi_{N-1}\right)}\right]}^{T},\;$$
where $\rho_{m}=\frac{2\pi Rf_{m}}{c}$ for m=1,…,M and $\Phi_{n}=\frac{2\pi n}{N}$ for n=1,…,N−1. To satisfy the strict requirements of the array deployment, a uniform linear array (ULA) is used for the destination. Hence, the array response vector at the secondary user is modeled as follows:(76)$$a_{r}^{m}\left(\varphi \right)=\frac{1}{\sqrt{N}}{\left[1,e^{j\frac{2\pi df_{m}}{c}\sin\left(\varphi \right)},\ldots ,e^{j\frac{2\pi \left(N-1\right)df_{m}}{c}\sin\left(\varphi \right)}\right]}^{T}.\;$$

The simulation parameters for visualizing the impact of UCA on the performance of the proposed technique are given in Table 7 (shown on the next page).

Figure 21 illustrates the spectral efficiency performance of the proposed approach, considering UCA at the source and relay nodes, while a uniform linear array (ULA) is deployed at each SU. The optimal fully digital beamforming is considered as the baseline for comparison. The number of antennas and the number of data streams are set as follows: $N_{t}=N_{r}=64$, $N_{d_{k}}=16$, K=4, and $N_{s}=8$. It is evident from the obtained results that the fully connected structure obtains approximately 83% of the optimal value, while the mixed architecture achieves nearly 76% of the baseline.

Figure 22 visualizes the impact on the performance of the proposed algorithm by increasing the number of antennas and the number of transmitted data streams. To generate simulation results, the various system parameters are given as $N_{t}=N_{r}=256$, $N_{d_{k}}=64$, K=4, and $N_{s}=16$. Numerical results show that the proposed method achieves approximately the same performance as obtained with a small number of antennas and data streams.

Note: It is worth noting that the proposed technique is different from the previous work [70], where a fully connected hybrid structure was considered at communicating nodes. The primary focus of this work is to derive an energy-efficient hybrid beamforming solution for CR-based relay-assisted MU-MIMO systems under frequency-selective channels. To achieve this goal, a combination of a fully connected architecture and a partially connected structure is employed at the relay node, while a partially connected structure is deployed at each SU. This structure is of great practical importance to achieve an energy-efficient hybrid beamforming solution while maximizing the achievable sum rate. All simulation results focus on comparing the performance of the proposed mixed hybrid architecture with the corresponding fully connected structure. Additionally, a sub-section is included in the simulation results that describes the energy consumption of the fully connected architecture, mixed structure, and fully digital precoding. Numerical results are also provided to show the energy efficiency of mixed structures in comparison to fully connected hybrid precoding and full-complexity digital beamforming. Finally, the spatial wideband effect and the impact of UCA on the performance of the presented technique are also included to show the effectiveness of the proposed approach through simulation results.

## 6. Conclusions

This paper investigates a hybrid wideband mm-wave transceiver for maximizing the sum rate in relay-assisted MU-MIMO CRNs. The underlying network architecture is of great practical importance in enhancing the efficiency of spectrum utilization. In addition, the overall performance can also be improved due to the cooperative communication paradigm. To achieve an energy-efficient solution for hybrid precoding, a combination of partially connected and fully connected structures is taken into consideration. A comparative study is also conducted between fully connected and mixed structures through computer simulations, which describes the better energy efficiency performance of the latter architecture compared to the former one. To reduce the complexity of the original optimization problem, an endeavor is made to decompose it into sub-problems. The focus of the proposed algorithm is to maximize the sum rate of SUs by keeping interference experienced by the PU within an acceptable limit. This task can be achieved by following a decoupled approach to address each sub-problem, where the RF and baseband processing components are derived separately. Moreover, the sum rate is maximized through the analog beamformers at different communicating nodes, and interference is minimized via frequency-dependent digital precoders and combiners. It is evident from the obtained results that the proposed method achieves performance close to that of its fully connected counterpart when different system parameters are changed over a wide range. A small performance degradation occurs in a gradual manner by increasing the channel estimation error in discrete steps. Simulation results also demonstrate that the mixed hybrid structure shows better energy efficiency when compared with its fully connected and full-complexity counterparts. This aspect of the suggested design is of prime significance. For promising future research directions: (1) The proposed design approach can be extended to multi-relay MIMO networks with other practical constraints due to the CR technology. These constraints mainly include interference from primary networks and power allocation to achieve different goals, such as minimizing the asymptotic outage probability, maximizing end-to-end throughput, and improving transmission performance. (2) It is also of great interest to investigate the performance of the proposed scheme by replacing the relay node with reconfigurable intelligent surfaces (RISs). They play an essential role in finding energy-efficient solutions with low complexity. Additionally, the deployment of a large number of RISs can achieve the same performance as relay networks.

## Figures and Tables

**Figure 1 sensors-24-03713-f001:**
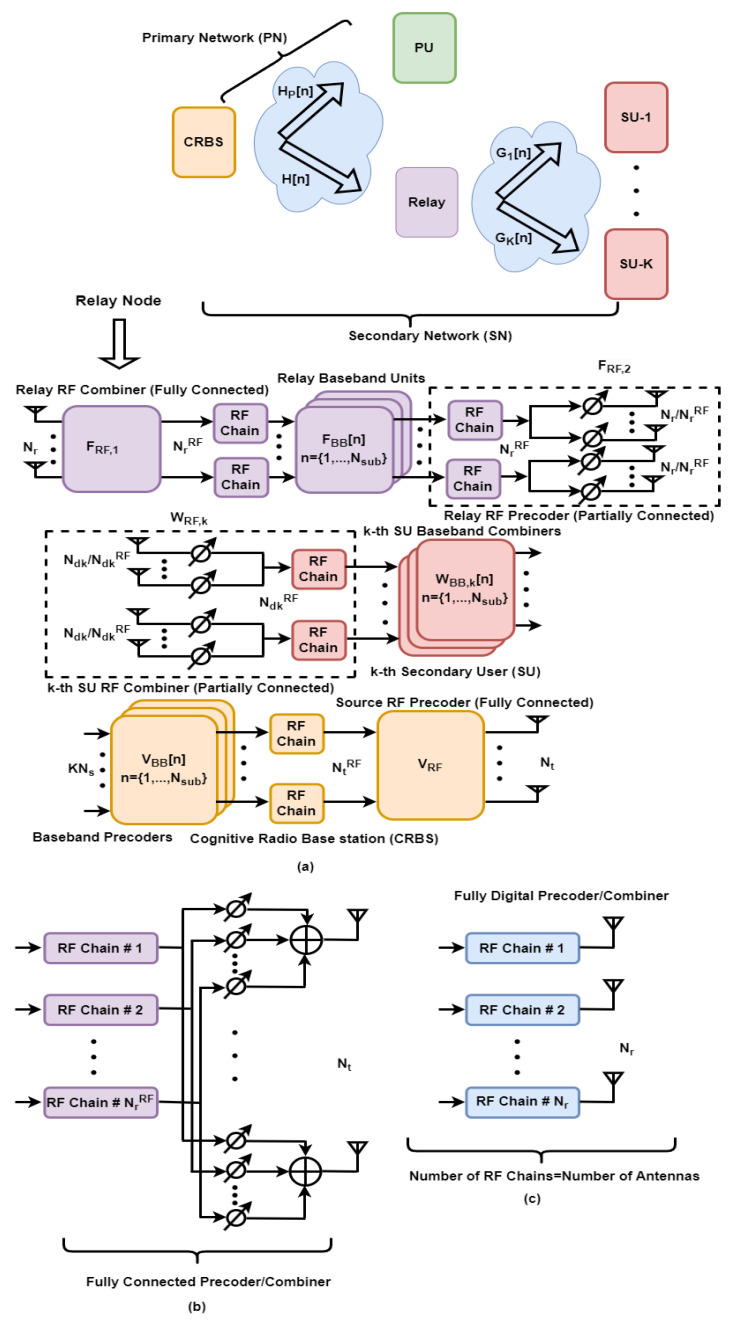
(**a**) System block diagram of a mixed-structure cognitive radio hybrid wideband transceiver in mm-wave relay-assisted MU-MIMO networks, (**b**) Fully Connected Structure, (**c**) Fully Digital system.

**Figure 2 sensors-24-03713-f002:**
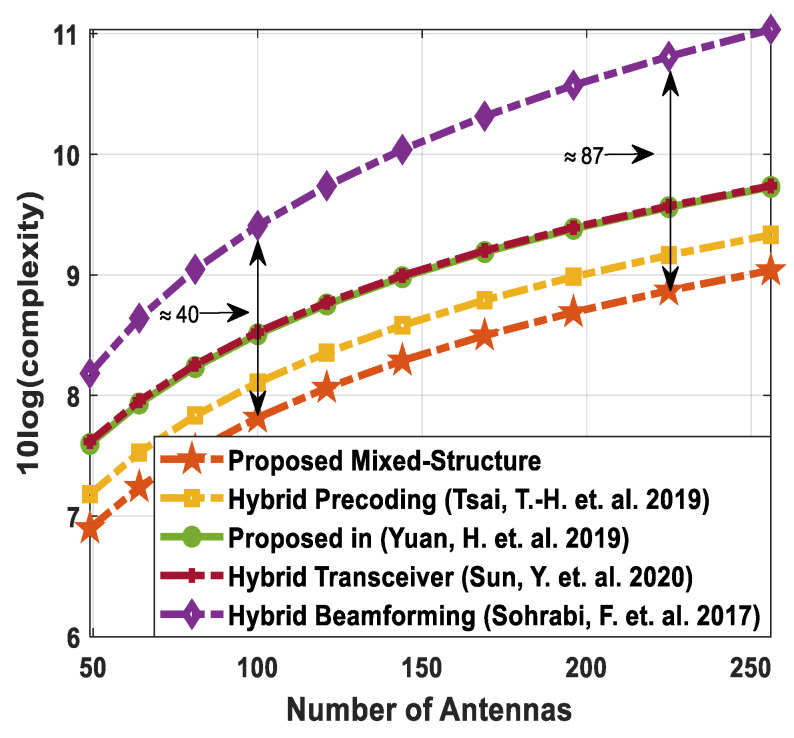
Complexity vs. number of antennas when $N_{sub}=64$, K=5, $N_{t}^{RF}=N_{r}^{RF}=KN_{s}$, $N_{d}^{RF}=N_{s}=5$, and $N_{d_{k}}=9$: Comparison of the proposed technique with other exiting hybrid precoding algorithms [41,43,47,50].

**Figure 3 sensors-24-03713-f003:**
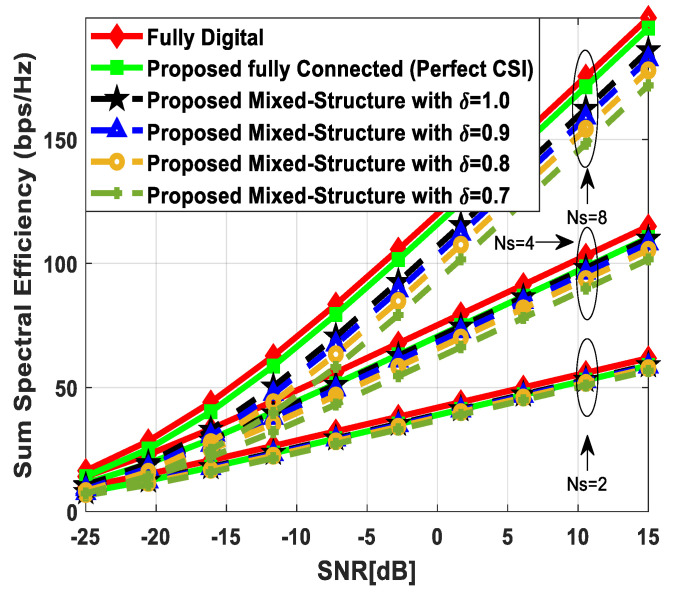
Sum spectral efficiency comparison of fully digital precoding, the proposed fully connected hybrid structure, and mixed hybrid architecture considering channel estimation error. The system parameters are set as $N_{t}=N_{r}=256$, $N_{d_{k}}=64$, K=4, $N_{s}=\left\{2,\;4,\;8\right\}$, and $N_{sub}=64$.

**Figure 4 sensors-24-03713-f004:**
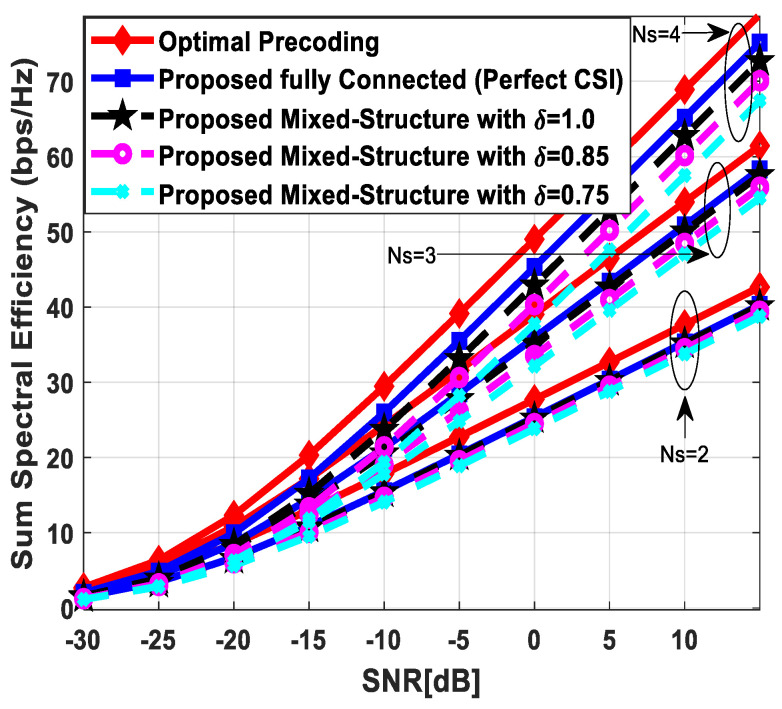
Sum spectral efficiency vs. SNR: Comparison of full-complexity digital precoding, the proposed fully connected hybrid architecture, and the mixed hybrid structure considering an imperfect channel with δ=0.85, 0.75. The system parameters are set as $N_{t}=N_{r}=144$, $N_{d_{k}}=36$, K=3, $N_{s}=\left\{2,\;3,\;4\right\}$, and $N_{sub}=64$ for conducting computer simulations.

**Figure 5 sensors-24-03713-f005:**
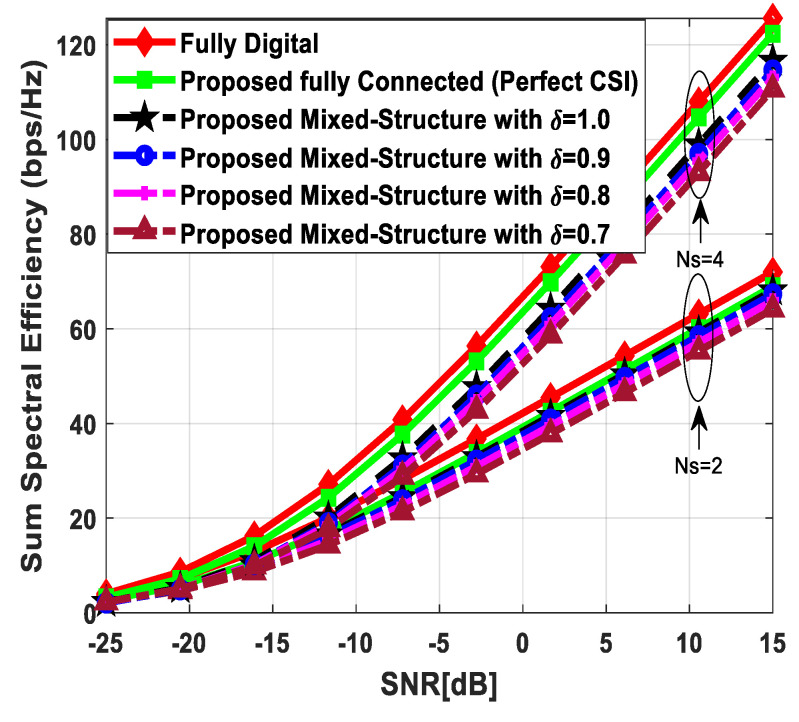
Sum spectral efficiency vs. SNR when $N_{t}=N_{r}=144$, $N_{d_{k}}=16$, *K* = 6, $N_{s}=\left\{2,\;4\right\}$, and $N_{sub}=64$. The estimated channel matrix is generated at δ=0.9, 0.8, 0.7.

**Figure 6 sensors-24-03713-f006:**
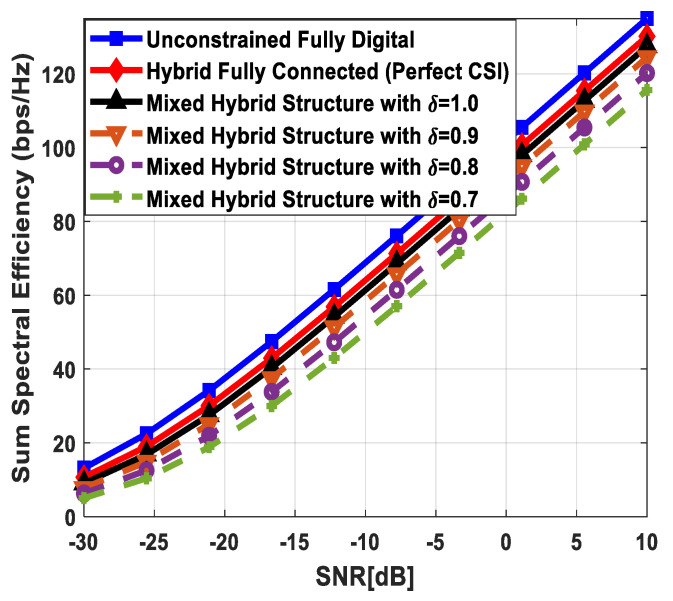
Achievable rate vs. SNR, considering URA: $N_{t}=N_{r}=200$, $N_{d_{k}}=50$, K=4, $N_{s}=5$, and $N_{sub}=64$. The estimated channel matrix is generated at δ=0.9, 0.8, 0.7.

**Figure 7 sensors-24-03713-f007:**
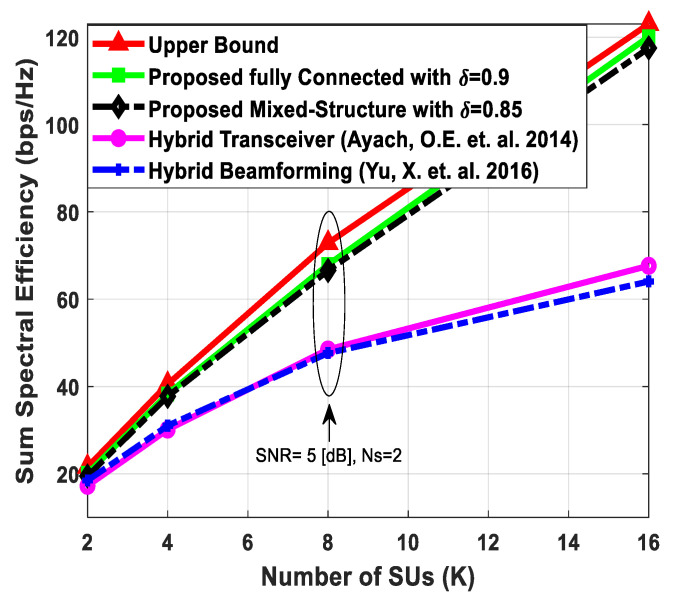
Sum spectral efficiency vs. number of SUs: Comparison of the proposed method with other hybrid precoding techniques. The system parameters are set as $N_{t}=N_{r}=256$, $N_{d_{k}}=16$, $N_{s}=2$, and $N_{sub}=64$. Impact on the rate of change of spectral efficiency as a function of the number of Sus [51,52].

**Figure 8 sensors-24-03713-f008:**
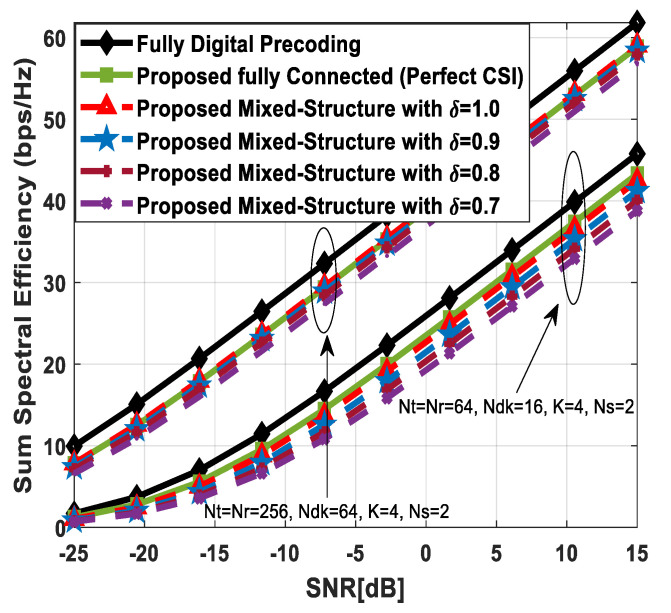
Sum spectral efficiency vs. SNR when the number of antennas at the source, relay node, and SUs is changed simultaneously, such as $N_{t}=N_{r}=64,\;256$, $N_{d_{k}}=16,\;64$. The constant parameters are set as K=4, $N_{s}=2$, and $N_{sub}=64$. Comparison of the proposed algorithm under perfect CSI and imperfect CSI at δ=0.9, 0.8, 0.7 with full-complexity digital precoding.

**Figure 9 sensors-24-03713-f009:**
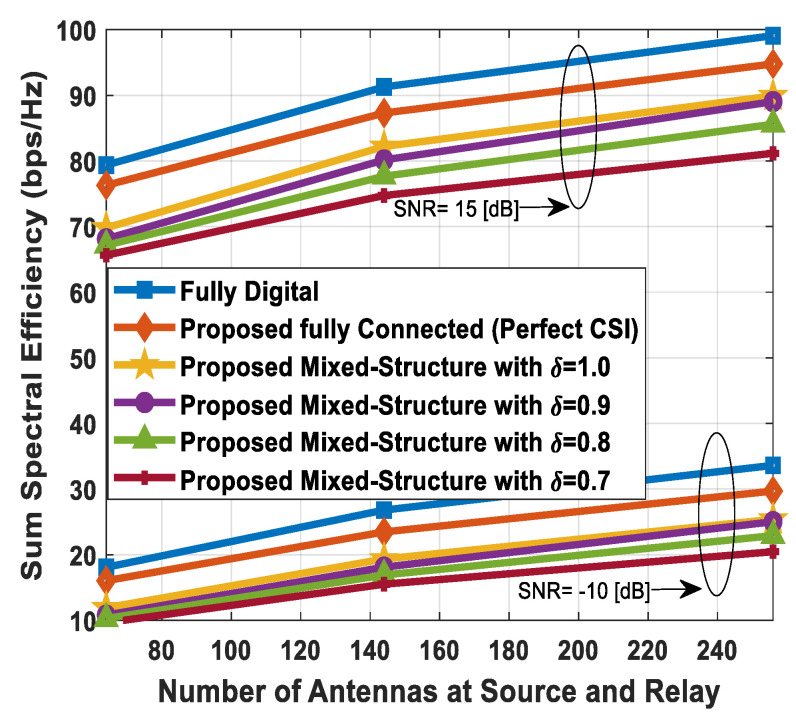
Sum spectral efficiency vs. number of antennas at the source and relay node when SNR= −10, 15 [dB], $N_{t}=N_{r}=64,\;144,\;256$, $N_{d_{k}}=16$, K=4, $N_{s}=4$, and $N_{sub}=64$. Comparison of the proposed technique with fully digital beamforming.

**Figure 10 sensors-24-03713-f010:**
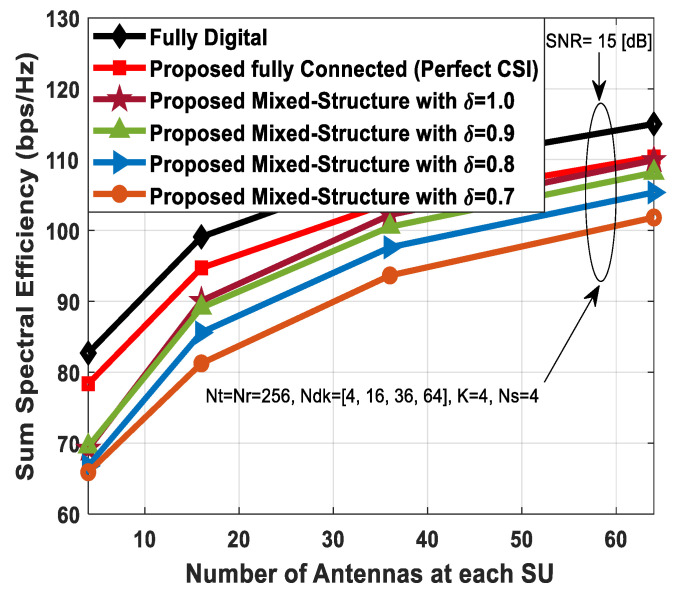
Sum spectral efficiency vs. number of antennas at the SUs when SNR = 15 [dB], $N_{t}=N_{r}=256$, $N_{d_{k}}=\left\{4,\;16,\;36,\;64\right\}$, K=4, $N_{s}=4$, and $N_{sub}=64$. Comparison of the proposed scheme under perfect CSI and imperfect CSI at δ=0.9, 0.8, 0.7 with full-complexity precoding.

**Figure 11 sensors-24-03713-f011:**
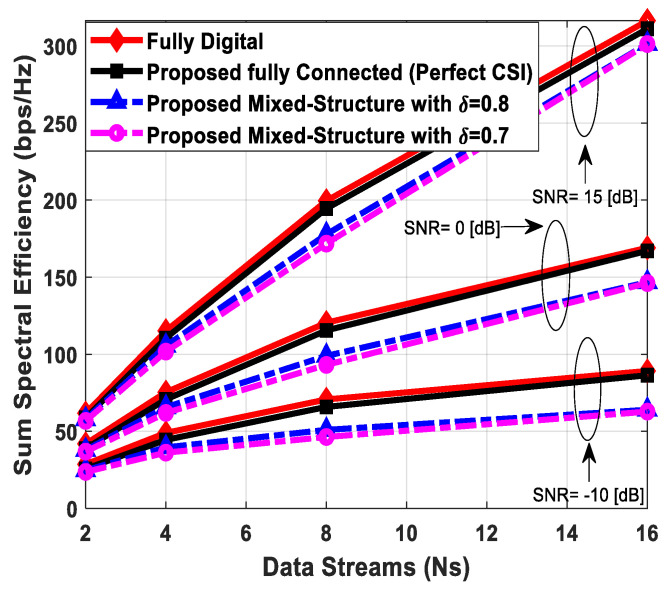
Sum spectral efficiency vs. number of data streams at SNR = −10, 0, 15 [dB]: Comparison of the proposed method with unconstrained fully digital precoding. The system parameters are set as follows: $N_{t}=N_{r}=256$, $N_{d_{k}}=64$, K=4, and $N_{sub}=64$.

**Figure 12 sensors-24-03713-f012:**
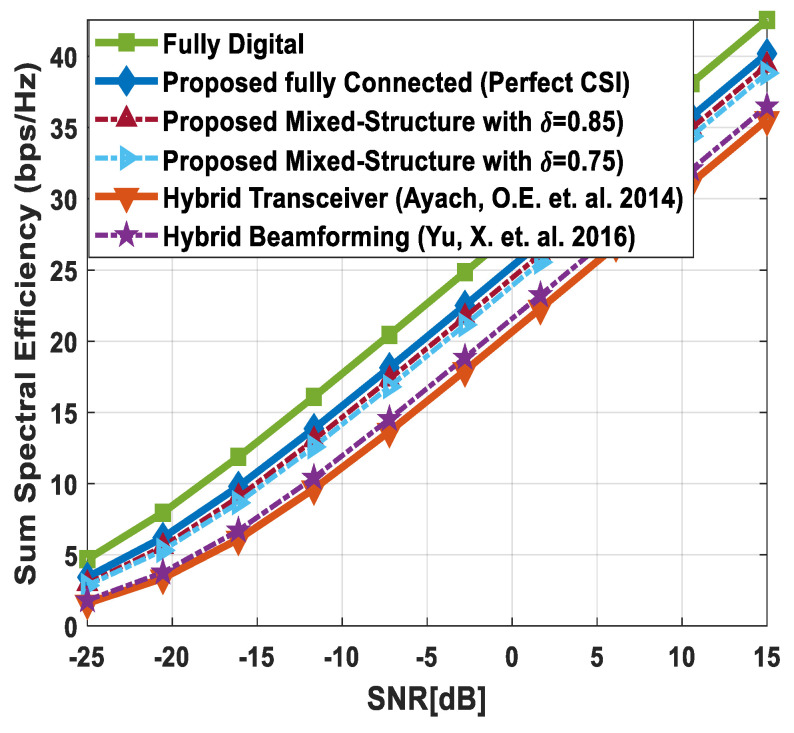
Sum spectral efficiency vs. SNR: Comparison of the proposed scheme with well-known hybrid processing techniques suggested in [51,52]. Numerical results are obtained by employing system parameters $N_{t}=N_{r}=144$, $N_{d_{k}}=36$, K=3, $N_{s}=2$, and $N_{sub}=64$.

**Figure 13 sensors-24-03713-f013:**
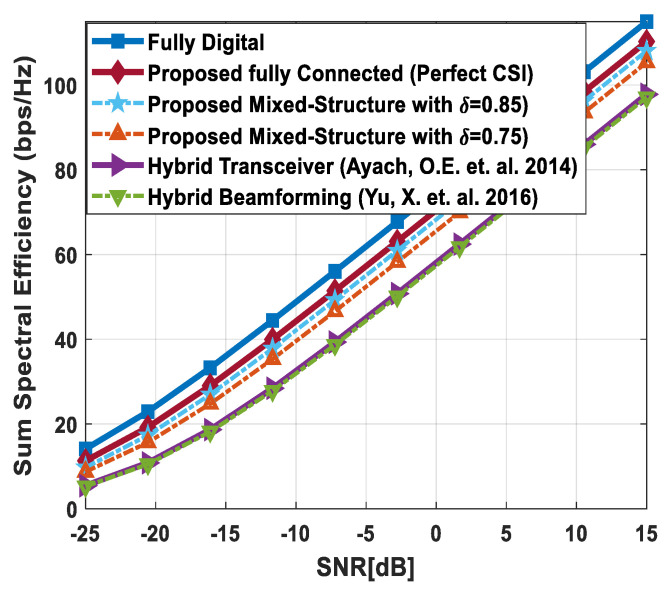
Sum spectral efficiency vs. SNR: Comparison between the proposed algorithm and hybrid precoding designs given in [51,52]. Simulation results are obtained using system parameters $N_{t}=N_{r}=256$, $N_{d_{k}}=64$, K=4, $N_{s}=4$, and $N_{sub}=64$.

**Figure 14 sensors-24-03713-f014:**
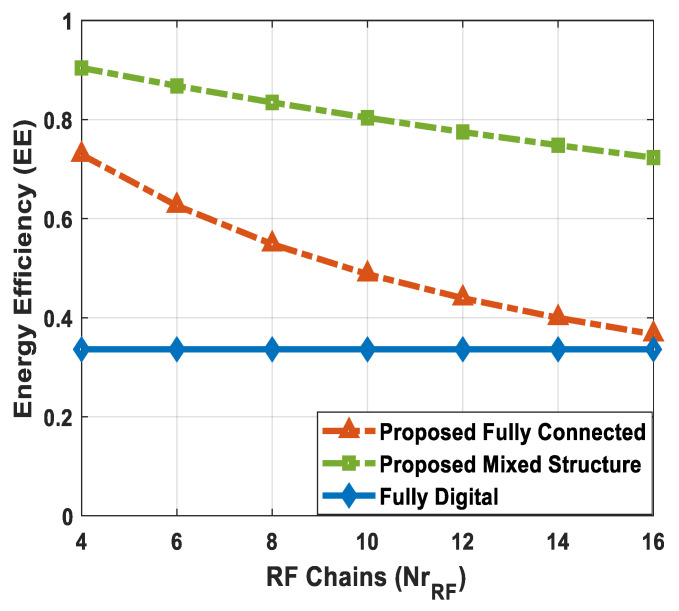
EE vs. number of RF chains at the relay node: Energy efficiency comparison of the proposed fully connected structure, mixed-structure, and full-complexity digital precoding. The system parameters are set as $N_{t}=N_{r}=100$, $N_{d_{k}}=25$, K=4, *N_s_* = 1, SNR=10 [dB], and $N_{sub}=64$.

**Figure 15 sensors-24-03713-f015:**
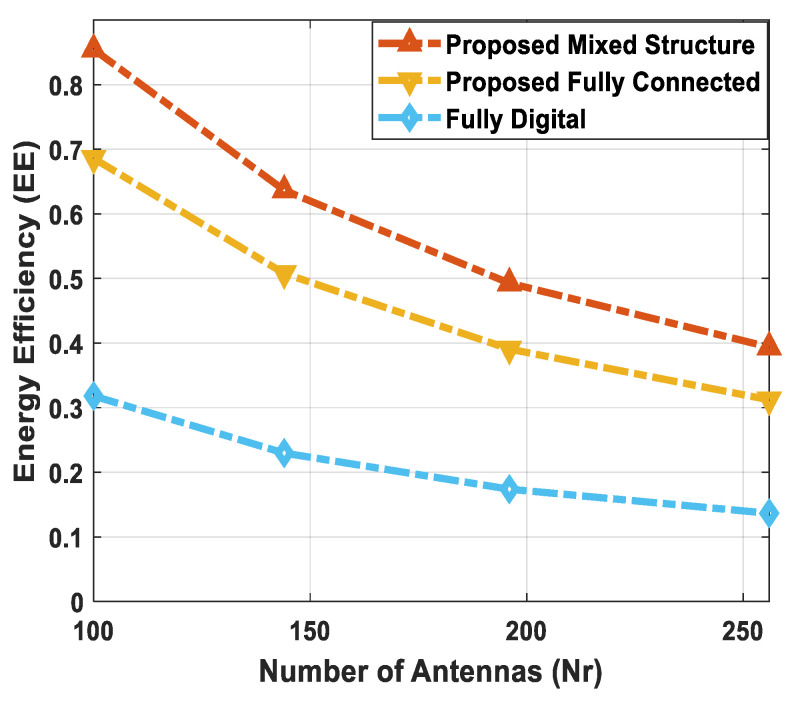
EE vs. number of antennas at the relay station: Comparison among the proposed fully connected architecture, mixed-structure, and fully digital beamforming. The system parameters are set as follows: $N_{t}=N_{r}=\left\{100,\;144,\;196,\;256\right\}$, $N_{d_{k}}=16$, K=4, $N_{s}=1$, SNR=10 [dB], and $N_{sub}=64$.

**Figure 16 sensors-24-03713-f016:**
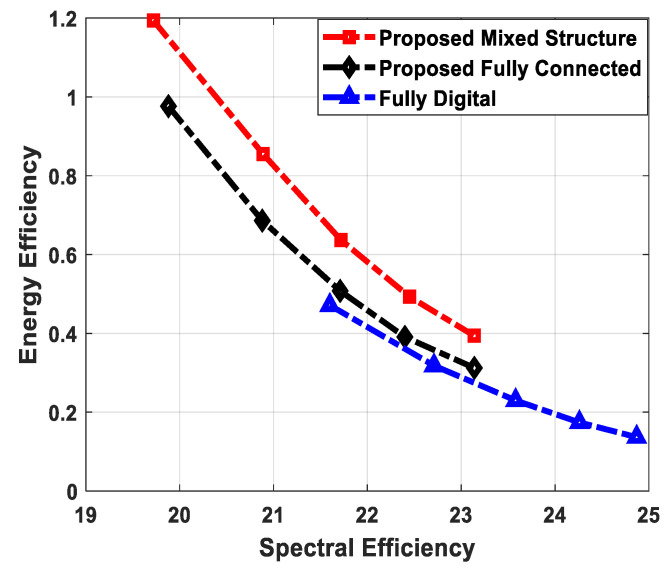
Sum spectral efficiency vs. EE with varying numbers of transmit antennas at the relay node $N_{t}=N_{r}\in \left\{64,\;100,\;144,\;196,\;256\right\}$ when $N_{d_{k}}=16$, K=4, $N_{s}=1$, SNR=10 [dB], and *N_sub_* = 64.

**Figure 17 sensors-24-03713-f017:**
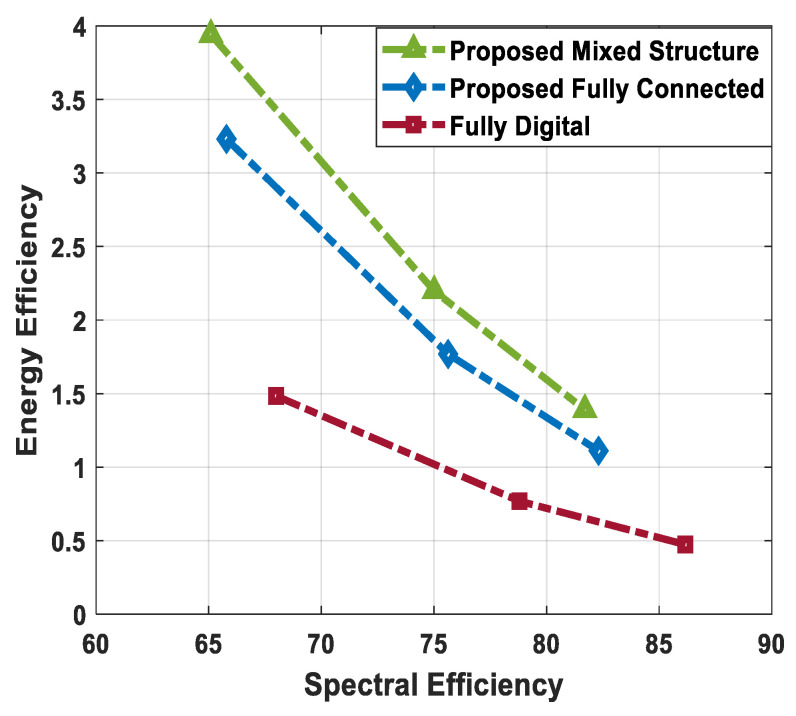
Sum spectral efficiency vs. EE by changing the number of transmit antennas at the relay station $N_{t}=N_{r}\in \left\{64,\;144,\;256\right\}$ when $N_{d_{k}}=16$, K=8, $N_{s}=2$, SNR=10 [dB], and $N_{sub}=64$.

**Figure 18 sensors-24-03713-f018:**
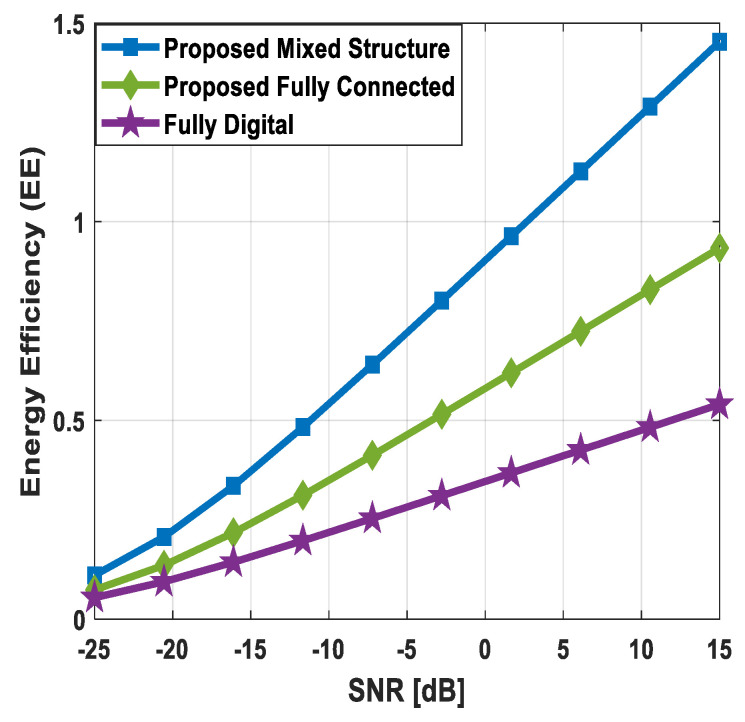
EE vs. SNR when $N_{t}=N_{r}=144$, $N_{d_{k}}=36$, K=4, $N_{s}=2$, and $N_{sub}=64$.

**Figure 19 sensors-24-03713-f019:**
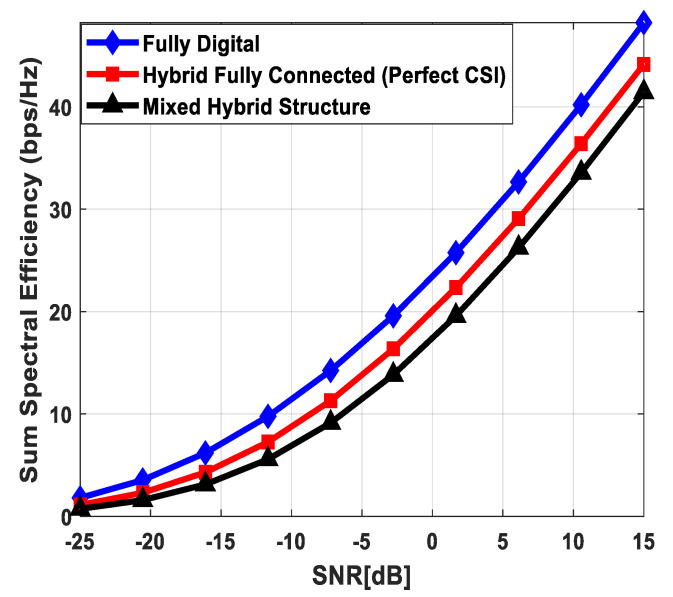
Rate vs. SNR considering spatial wideband effect: $N_{t}=N_{r}=36$, $N_{d_{k}}=9$, K=4, $N_{s}=3$, and $N_{sub}=64$.

**Figure 20 sensors-24-03713-f020:**
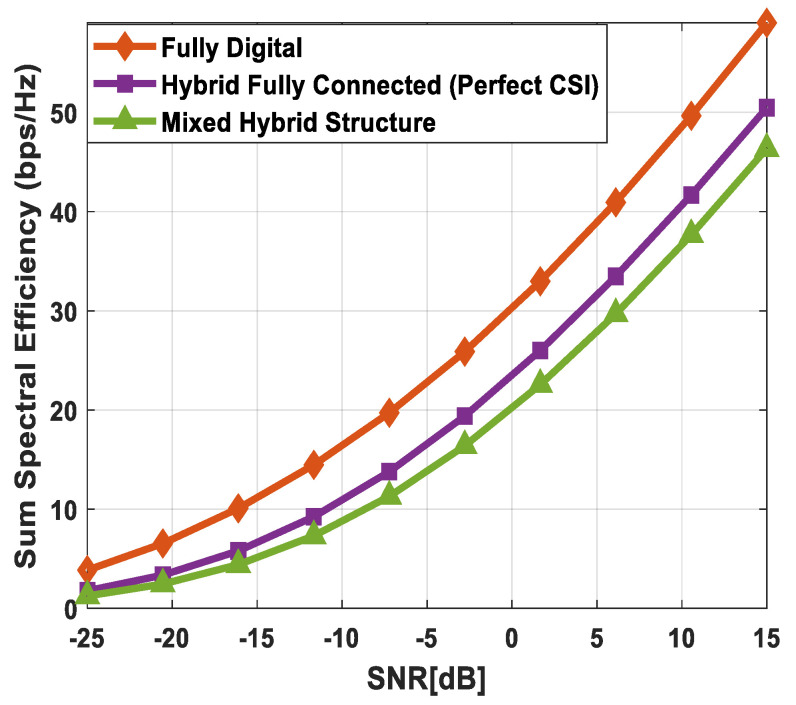
Rate vs. SNR considering spatial wideband effect: $N_{t}=N_{r}=64$, $N_{d_{k}}=16$, K=4, $N_{s}=4$, and $N_{sub}=64$.

**Figure 21 sensors-24-03713-f021:**
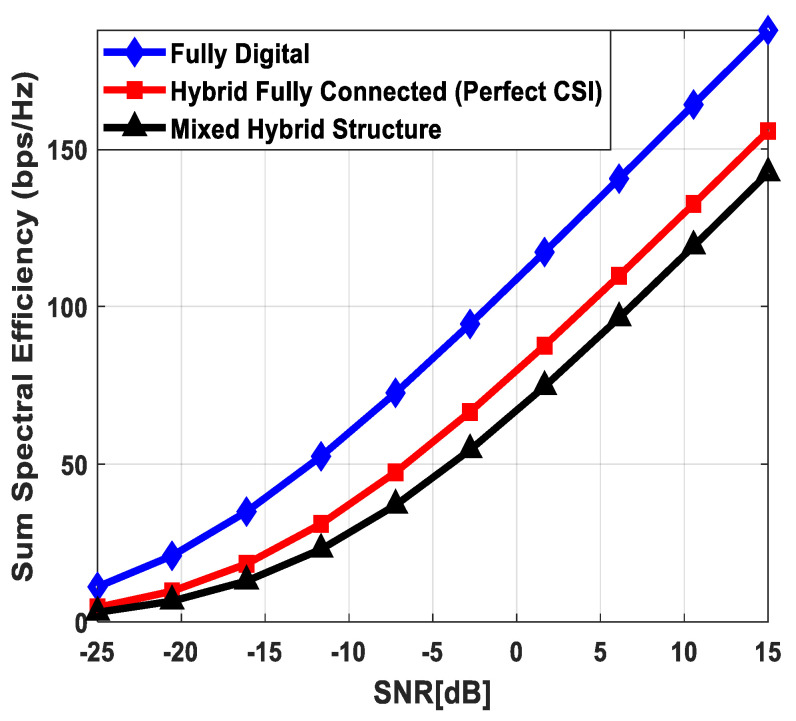
Rate vs. SNR considering the impact of UCA: $N_{t}=N_{r}=64$, $N_{d_{k}}=16$, K=4, $N_{s}=8$, and $N_{sub}=64$.

**Figure 22 sensors-24-03713-f022:**
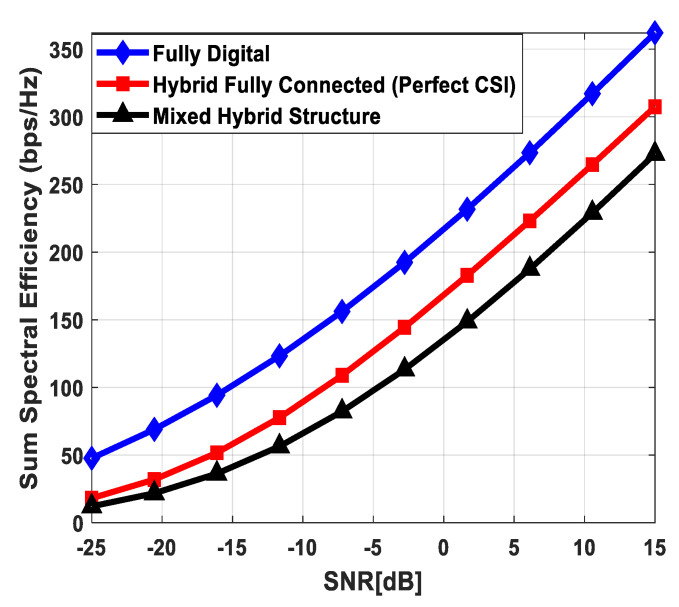
Rate vs. SNR considering the impact of UCA: $N_{t}=N_{r}=256$, $N_{d_{k}}=64$, K=4, $N_{s}=16$, and $N_{sub}=64$.

**Table 1 sensors-24-03713-t001:** Contrasting the proposed scheme with the existing relay-based hybrid beamforming for mm-wave MU-MIMO networks.

	[54]	[66]	[67]	[68]	[69]	[70]	Proposed
Relay-assisted mm-wave MU-MIMO system	✓	✓	✓	✓	✓	✓	✓
Frequency-selective channel	✕	✕	✕	✕	✕	✓	✓
Multiple data streams per user	✓	✓	✓	✓	✓	✓	✓
Minimum number of RF chains	✕	✕	✕	✕	✕	✓	✓
Support of CR technology	✕	✕	✕	✕	✕	✓	✓
Mixed hybrid structure	✕	✕	✕	✓	✕	✕	✓
Numerical results with imperfect CSI	✕	✕	✕	✕	✕	✓	✓

**Table 2 sensors-24-03713-t002:** Symbolic representation.

Symbol	Definition
$N_{t}/N_{r}/N_{d_{k}}$	Number of antennas at the source/relay node/k-th SU
$N_{t}^{RF}/N_{r}^{RF}/N_{d_{k}}^{RF}$	Number of RF chains at the source/relay node/k-th SU
$s_{k}\left[n\right]/s\left[n\right]$	Number of data streams transmitted to the k-th SU/K SUs
x[n]	Source-transmitted signal after hybrid precoding
$V_{RF}/V_{BB}\left[n\right]$	Common analog RF beamformer/frequency-selective digital baseband precoder at the source
$P_{s}$	Maximum allowable power at the source
$y_{r}\left[n\right]$	Signal received at the relay node
$d_{1}\left[n\right]$	Interference experienced by the PU due to the source-transmitted signal
$y_{r_{1}}\left[n\right]$	Baseband signal received at the output of relay hybrid combiner
$F_{RF,1}/F_{RF,2}$	Common RF combiner/precoder at the relay station
$F_{BB,1}\left[n\right]/F_{BB,2}\left[n\right]$	Frequency-dependent baseband combiner/precoder at the relay
$F_{1}\left[n\right]/F_{2}\left[n\right]$	Relay hybrid combiner/precoder
$F\left[n\right]/F_{BB}\left[n\right]$	Relay hybrid filter/combined baseband processing component at the relay
$y_{r_{2}}\left[n\right]$	Relay transmitted signal after hybrid beamforming
$d_{2}\left[n\right]$	Interference experienced by the PU due to the relay-transmitted signal
$H\left[n\right]/H_{PU}\left[n\right]/G_{d_{k}}\left[n\right]$	Channel matrix from the source to the relay in frequency-domain/source to the PU/relay to the k-th SU
$n_{r}\left[n\right]/z_{k}\left[n\right]$	Noise vector at the relay/k-th SU
$y_{d_{k}}^{+}\left[n\right]/y_{d_{k}}\left[n\right]$	Signal received at the k-th SU without hybrid processing/with hybrid processing
$W_{RF,k}/W_{BB,k}\left[n\right]$	Frequency-independent RF combiner/frequency-dependent baseband combiner at the k-th SU
$H_{eq}\left[n\right]=F_{RF,1}^{H}H\left[n\right]V_{RF}$	Baseband equivalent channel from the source to the relay
$G_{eq,k}\left[n\right]=W_{RF,k}^{H}G_{d_{k}}\left[n\right]F_{RF,2}$	Baseband equivalent channel from the relay to the k-th SU
$C_{k}\left[n\right]/R_{avg}$	Capacity of the *k*-th SU at the *n*-th sub-carrier/average over $N_{sub}$ carriers
$F_{2}^{k}\left[n\right]=F_{RF,2}F_{BB,2}^{k}\left[n\right]$	Relay hybrid precoder corresponding to the k-th SU

**Table 3 sensors-24-03713-t003:** Complexity of the proposed design and the other hybrid beamforming algorithms.

Algorithms	Complexity
Proposed, [70]	$N_{t}^{2}N_{r}+{N_{t}}^{2}N_{r}N_{sub}+N_{d_{k}}^{2}N_{r}+N_{d_{k}}^{2}N_{r}N_{sub}+{\left(N_{r}^{RF}\right)}^{3}+{\left(N_{d_{k}}^{RF}\right)}^{3}$
Hybrid Precoding [43]	$N_{s}^{4}+N_{s}^{3}\left(N_{t}+N_{r}\right)+{\left(N_{t}N_{r}N_{s}\right)}^{2}+N_{sub}\left(N_{s}^{3}+N_{t}^{2}N_{r}+N_{r}^{2}N_{s}\right)$
Hybrid Beamforming [47]	$N_{sub}\left(N_{s}^{3}+N_{t}^{2}N_{r}+N_{t}\left({N_{s}}^{2}+{N_{r}}^{2}\right)\right)+N_{s}\left({N_{t}}^{2}+{N_{r}}^{2}\right)$
Hybrid Transceiver [41]	$O\left(KN_{sub}N_{r}N_{t}^{2}+KN_{sub}N_{s}N_{t}^{2}\right)$
Algorithm [50]	$O\left(KN_{sub}N_{r}N_{t}^{2}+KN_{sub}N_{s}N_{t}N_{t}^{RF}\right)$

**Table 4 sensors-24-03713-t004:** System parameters to generate numerical results.

Parameters	Values
Number of data streams	$N_{s}=1\sim 16$
Number of RF chains	$N_{t}^{RF}=N_{r}^{RF}=KN_{s},N_{d_{k}}^{RF}=N_{s}$
Number of antennas	$N_{t}=N_{r}=49\sim 256,N_{d_{k}}=4\sim 64$
Number of data transmission paths	$N_{cl}=5,\;N_{ray}=10$
Number of frequency sub-carriers	$N_{sub}=64$
Carrier frequency	$f_{c}=28\;\mathrm{GHz}$
Number of secondary users	K=2~16

**Table 5 sensors-24-03713-t005:** Energy consumption of different precoding techniques.

Architecture	Energy Consumption at the Relay Node
$P_{FD}$	$2P_{BB}+N_{r}\left(2P_{RF}+P_{PA}+P_{LNA}+P_{ADC}+P_{DAC}\right)$
$P_{FC}$	$2P_{BB}+2N_{r}^{RF}P_{RF}+N_{r}\left(P_{PA}+P_{LNA}\right)+2N_{r}N_{r}^{RF}P_{PS}+N_{r}^{RF}\left(P_{ADC}+P_{DAC}\right)$
$P_{PC}$	$2P_{BB}+2N_{r}^{RF}P_{RF}+N_{r}\left(P_{PA}+P_{LNA}\right)+2N_{r}P_{PS}+N_{r}^{RF}\left(P_{ADC}+P_{DAC}\right)$

**Table 6 sensors-24-03713-t006:** Simulation parameters for spatial wideband effect.

Parameters	Values
Number of data streams	$N_{s}=3,\;4$
Number of SUs	K=4
Number of sub-carriers	$N_{sub}=64$
Number of antennas at the source, relay node, and each SU	$N_{t}=N_{r}=36,\;64$ $,\;N_{d_{k}}=9,\;16$
Number of clusters; number of rays/cluster	$N_{cl}=4,\;N_{ray}=5$
Maximum time delay	$\tau_{max}=20\;\mathrm{ns}$
Central frequency	$f_{c}=30\;\mathrm{GHz}$
Bandwidth	$f_{s}=3\;\mathrm{GHz}$

**Table 7 sensors-24-03713-t007:** Simulation parameters for the impact of UCA.

Parameters	Values
Number of data streams	$N_{s}=8,\;16$
Number of SUs	K=4
Number of sub-carriers	$N_{sub}=64$
Number of antennas at the source, relay node, and each SU	$N_{t}=N_{r}=64,\;256$, $N_{d_{k}}=16,\;64$
Number of propagation paths	L=20
Maximum time delay	$\tau_{max}=25\;\mathrm{ns}$
Central frequency	$f_{c}=30\;\mathrm{GHz}$
Bandwidth	$f_{s}=3\;\mathrm{GHz}$

## Data Availability

Data can be generated using the mathematical expressions given in the manuscript to generate simulations.

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
