# Peer review of "Spectral Efficiency Maximization for Mixed-Structure Cognitive Radio Hybrid Wideband Millimeter-Wave Transceivers in Relay-Assisted Multi-User Multiple-Input Multiple-Output Systems"

_sensors, 2024, doi:10.3390/s24123713_

Round 1
Reviewer 1 Report
Comments and Suggestions for Authors
The authors of this paper proposes a cognitive radio network (CRN)-based hybrid wideband precoding for maximizing spectral efficiency in millimeter-wave relay-assisted multi-user (MU) multiple-input multiple-output (MIMO) systems. Both analytical and simulation results were provided. The paper is well written and has technical quality. However, there are few areas that require little modification to enhance the quality of the paper.
Introduction
· This is well written. However, few sentences that may require additional citations such as line 55-56 and many others.
· The motivation of the paper is very clear, but there is a need to explicitly, inform the reader the gap this paper addressed references to the existing works
· Contribution of the paper is not highlighted also.
Related Works
· The review of related works is well coordinated and written. Extensive review was provided, but the main weakness of this section is that the bibliography is bit outdated. Most of the latest of the papers reviewed was 2020.
· There are quite lot of recent research works that could be easily reviewed and added to the bibliography list of the paper, I can cite some papers from reputable databases within 2023-2024:
1. Wang, Q., Li, P., Rocca, P., Li, R., Tan, G., Hu, N.,... Xu, W. (2023). Interval-Based Tolerance Analysis Method for Petal Reflector Antenna With Random Surface and Deployment Errors. IEEE Transactions on Antennas and Propagation, 71(11), 8556-8569. doi: 10.1109/TAP.2023.3314097
2. Xu, G., Zhang, Q., Song, Z., & Ai, B. (2023). Relay-Assisted Deep Space Optical Communication System Over Coronal Fading Channels. IEEE Transactions on Aerospace and Electronic Systems, 59(6), 8297-8312. doi: 10.1109/TAES.2023.3301463
3. Yin, Y., Guo, Y., Su, Q., & Wang, Z. (2022). Task Allocation of Multiple Unmanned Aerial Vehicles Based on Deep Transfer Reinforcement Learning. Drones, 6(8), 215. doi: 10.3390/drones6080215
4. Zhou, P., Zheng, P., Qi, J., Li, C., Lee, H., Duan, A.,... Navarro-Alarcon, D. (2024). Reactive human–robot collaborative manipulation of deformable linear objects using a new topological latent control model. Robotics and Computer-Integrated Manufacturing, 88, 102727. doi: https://doi.org/10.1016/j.rcim.2024.102727
5. Yang, M., Cai, C., Wang, D., Wu, Q., Liu, Z.,... Wang, Y. (2024). Symmetric Differential Demodulation-Based Heterodyne Laser Interferometry Used for Wide Frequency-Band Vibration Calibration. IEEE Transactions on Industrial Electronics, 71(7), 8132-8140. doi: 10.1109/TIE.2023.3299015
6. Gao, N., Liu, J., Deng, J., Chen, D., Huang, Q.,... Pan, G. (2024). Design and performance of ultra-broadband composite meta-absorber in the 200Hz-20kHz range. Journal of Sound and Vibration, 574, 118229. doi: https://doi.org/10.1016/j.jsv.2023.118229
7. Zhao, L., Xu, H., Qu, S., Wei, Z., & Liu, Y. (2024). Joint Trajectory and Communication Design for UAV-Assisted Symbiotic Radio Networks. IEEE Transactions on Vehicular Technology. doi: 10.1109/TVT.2024.3356587
8. Yang, Y., Zhang, Z., Zhou, Y., Wang, C., & Zhu, H. (2023). Design of a Simultaneous Information and Power Transfer System Based on a Modulating Feature of Magnetron. IEEE Transactions on Microwave Theory and Techniques, 71(2), 907-915. doi: 10.1109/TMTT.2022.3205612
9. Ning, Y., Zhu, S., Chu, H., Zou, Q., Zhang, C., Li, J.,... Li, G. (2024). 1-bit Low-Cost Electronically Reconfigurable Reflectarray and Phased Array Based on p-i-n Diodes for Dynamic Beam Scanning. IEEE Transactions on Antennas and Propagation, 72(2), 2007-2012. doi: 10.1109/TAP.2023.3325650
10. Xiao, N., Wang, Y., Chen, L., Wang, G., Wen, Y.,... Li, P. (2023). Low-Frequency Dual-Driven Magnetoelectric Antennas With Enhanced Transmission Efficiency and Broad Bandwidth. IEEE Antennas and Wireless Propagation Letters, 22(1), 34-38. doi: 10.1109/LAWP.2022.3201070
11. Li, Y., Luo, Y., Wu, X., Shi, Z., Ma, S.,... Yang, G. (2024). Variational Bayesian Learning Based Localization and Channel Reconstruction in RIS-aided Systems. IEEE Transactions on Wireless Communications. doi: 10.1109/TWC.2024.3380903
12. Chen, J., Wang, X., Fang, Z., Jiang, C., Gao, M.,... Xu, Y. (2024). A Real-Time Spoofing Detection Method Using Three Low-Cost Antennas in Satellite Navigation. Electronics, 13(6), 1134. doi: https://doi.org/10.3390/electronics13061134
13. Zhang, H., Xu, Y., Luo, R., & Mao, Y. (2023). Fast GNSS acquisition algorithm based on SFFT with high noise immunity. China Communications, 20(5), 70-83. doi: 10.23919/JCC.2023.00.006
14. Wang, R., Gu, Q., Lu, S., Tian, J., Yin, Z., Yin, L.,... Zheng, W. (2024). FI-NPI: Exploring Optimal Control in Parallel Platform Systems. Electronics , 13(7), 1168. doi: https://doi.org/10.3390/electronics13071168
15. Huang, X., Zhang, X., Zhou, L., Xu, J., & Mao, J. (2023). Low-Loss Self-Packaged Ka-Band LTCC Filter Using Artificial Multimode SIW Resonator. IEEE Transactions on Circuits and Systems II: Express Briefs, 70(2), 451-455. doi: 10.1109/TCSII.2022.3173712
· Section 2.2
Line 393, citation is required for Saleh-Valenzuela model.
· Figure 2.0, the symbol should be different for each curve to enable clarity when printed in black and white.
· Figure 3.0, as above
· Check all other figures.
· Most of the derivations could be taken to Appendix.
Author Response
Manuscript ID: sensors-2988477
Type of manuscript: Article
Title: " Spectral Efficiency Maximization for Mixed-Structure Cognitive Radio
Hybrid Wideband Millimeter-Wave Transceiver in Relay-Assisted Multi-User MIMO
Systems "
Dear Ms. Rosy Zhan,
Associate Editor, MDPI Sensors,
We are very grateful to you for coordinating the review process of the manuscript mentioned above in an excellent manner. The authors would like to express their gratitude to the Associate Editor and all the Reviewers for their helpful and highly constructive remarks to improve and make the submitted manuscript technically sound.
Furthermore, the authors would like to thank the Associate Editor and reviewers for allowing a resubmission of the manuscript with an opportunity to address the recommendations, suggestions, and guidelines made by the reviewers.
We are uploading (a) our point-by-point response to the comments (below) (Response to Reviewers), (b) an updated manuscript with red text indicating changes made.
Thank you,
Yours sincerely,
Hafiz Muhammad Tahir Mustafa et al.

Reviewer 2 Report
Comments and Suggestions for Authors
This paper, which was analyzed in consideration of SE and EE in a CR application system using mm-wave OFDM, is recognized to be of great significance. However, it is necessary to answer some of the corrections and questions below.
<Contents to be revised>
1. Make sure that the following two are shown in Figure 1.
- Illustrate the difference between the structure of fully digital, hybrid transceiver, and hybrid beamforming, which are mentioned as other technologies, and the structure proposed in the paper so that it is easy to understand.
- Illustrate the fully connected structure and mixed structure presented in the proposed structure in an easy-to-understand manner too.
2. Requires consistent creation of .or, which is the termination and connection of all of the formula.
3. The formulas of line 478 and 479 need to be modified or organized.
4. Need to elaborate on the meaning of delta values defined in Figure 3 as channel estimation errors
<Question Contents>
1. For CR, low power operation is the key. It is necessary to further explain the purpose of studying OFDM, which consumes a lot of power, in CR application.
2. As mentioned in line 772, CR is important for cost-effectiveness, energy-efficiency, and low-complexity. The OFDM model has the opposite effect. Further explanation is needed for the purpose of research by applying mm-wave OFDM.
3. Only loss and antenna correlation are mentioned in the mmWave's channel model. In an environment where multiple CR-SUs are used, multi-path fading consideration seems important, and this consideration is also necessary.
4. Explain why you used the Saleh-Valenzuela model.
5. In Figure 2, why does the proposed technology improve complexity as the number of antennas increases?
6. I don't see the consistency of the results of Figure 3 and Figure 10. Need to supplement or explain this.
7. The SE result value in Figure 12 shows an excessive value compared to the other figures. Need an explanation for this.
8. Need to explain the difference of structure between RF Chains in Figure 13 and Number of antennas in Figure 14.
9. In Figure 13, despite the increase in RF chains, what is the meaning of the result of a constant EE value in a fully digital structure?
10. Below is the question for Figure 15.
- What does EE mean by 1 or more?
- Why is the EE value of the Fullly Digital structure not defined at the low SE value?
- Need to explain the difference with incorrect SE range compared to Fig. 16.
- In Figures 15 and 16, the higher the SE value, the lower the EE value. However, when the SE value in Figure 15 is 20, the EE value is 1.2, but Figure 16 shows the EE value of 4 even though the SE value is 65. Need to explain the reason for this.
Comments on the Quality of English Language..
Author Response

(The authors gave the same response as above.)

Reviewer 3 Report
Comments and Suggestions for Authors
The authors wrote an article to maximize the spectral efficiency of relay assisted multi-user MIMO systems for mm-wave transceiver.
Can the authors please justify the difference between the two published papers with the titles "Hybrid Wideband Millimeter Wave Transceiver for Single-User Multi-Relay MIMO Systems" and "Hybrid Wideband Beamforming for Sum Spectral Efficiency Maximization in Millimeter-Wave Relay-Assisted Multiuser MIMO Cognitive Radio Networks" and a proposed one? Instead of changing the parameters in equations and the axis range in figures/ graphs.
If the proposed one is an extension of the mentioned above, then please refer to these.
This article plagiarized more than 20% from a single source. Justify
Author Response

(The authors gave the same response as above.)

Reviewer 4 Report
Comments and Suggestions for Authors
This paper examines the performance of Cognitive Radio Networks (CRNs) with millimeter-wave communications, a topic that aligns well with the scope of the Sensors journal. However, several weaknesses have been identified:
-
The authors overlook interference from primary networks, a crucial consideration in cognitive radio networks. Additionally, they oversimplify intra-interference in secondary networks by treating it as noise, which significantly simplifies the mathematical framework, design, and optimization.
-
The authors only consider a uniform square planar antenna array. It would be valuable to investigate how system performance is impacted when the antenna array is circularly arranged.
-
In the simulation section, key parameters such as the transmit power of secondary transmitters (cognitive base stations), transmission distance, and beamforming gain are missing, particularly crucial in millimeter-wave communications.
-
The reviewer suggests exploring the system's performance at different carrier frequencies within the mm-wave bands, such as 70 GHz, 90 GHz, etc.
-
All equations not proposed by the authors should be properly cited to the original work.
-
The authors need to clarify how their proposed method surpasses existing state-of-the-art techniques and provide a careful justification for their approach.
-
In the conclusion section, more extensive discussions about potential extensions of the proposed approach would be beneficial.
-
Keywords should be presented in alphabetical order to enhance organization.
-
The manuscript should be thoroughly proofread to eliminate any remaining typos or errors.
-
If feasible, DOIs should be added for all references, and the manuscript should strictly adhere to the conference template.
The English is okay
Author Response

(The authors gave the same response as above.)

Round 2
Reviewer 2 Report
Comments and Suggestions for Authors
It was confirmed that the paper was revised appropriately to reflect opinions.
Reviewer 3 Report
Comments and Suggestions for Authors
Thanks for addressing my comments, now the paper is recommended for publication.
Reviewer 4 Report
Comments and Suggestions for Authors
I have no more comments.
Comments on the Quality of English LanguageEnglish is so so